# A Closer Look at Prototype Classifier for Few-shot Image Classification

## Abstract

The prototypical network is a prototype classifier based on meta-learning and is widely used for few-shot learning because it classifies unseen examples by constructing class-specific prototypes without adjusting hyper-parameters during meta-testing. Interestingly, recent research has attracted a lot of attention, showing that training a new linear classifier, which does not use a meta-learning algorithm, performs comparably with the prototypical network. However, the training of a new linear classifier requires the retraining of the classifier every time a new class appears. In this paper, we analyze how a prototype classifier works equally well without training a new linear classifier and meta-learning. We experimentally find that directly using the feature vector extracted using standard pre-trained models to construct a prototype classifier in meta-testing does not perform as well as the prototypical network and training a new linear classifiers and feature vectors of pre-trained models. Thus, we derive a novel generalization bound for the prototypical network and show that focusing on the variance of the norm of a feature vector can improve performance. We experimentally investigate several normalization methods for minimizing the variance of the norm and find that the same performance can be obtained by using the L2 normalization and embedding space transformation without training a new classifier or meta-learning.

## 1 Introduction

Few-shot learning is used to adapt quickly to new classes with low annotation cost. Meta-learning is a standard training procedure to tackle the few-shot learning problem and *Prototypical Network* (Snell et al., 2017) a.k.a ProtoNet is a widely used meta-learning algorithm for few-shot learning. In ProtoNet, we use a prototype classifier based on meta-learning to predict the classes of unobserved objects by constructing class-specific prototypes without adjusting the hyper-parameters during meta-testing.

ProtoNet has the following advantages. (1) Since the nearest neighbor method is applied on query data and class prototypes during the meta-test phase, no hyper-parameters are required in meta-test phase. (2) Since the number of data for few-shot is small, the inference time is almost negligible. (3) The classifiers can quickly adapt to new environments because they do not have to be re-trained for the support set when new classes appear. The generalization bound of ProtoNet in relation to the number of shots in a support set has been studied (Cao et al., 2020). The bound suggests that the performance of ProtoNet depends on the ratio of the between-class variance to the within-class variance of features of a support set extracted using the meta-trained model.

There have been studies on training a new linear classifier on the features extracted using a pre-trained model without meta-learning, which can perform comparably with the meta-learned models (Chen et al., 2019; Tian et al., 2020). We call this approach as linear-evaluation-based approach. In these studies, the models are trained with the standard classification problem, i.e., models are trained with cross-entropy loss after linear projection from the embedding space to the class-probability space. The linear-evaluation-based approach has the following advantages over meta-learning. (1) Training converges faster than meta-learning. (2) Implementation is simpler. (3) Meta-learning decreases in performance if the number of shots does not match between meta-training and meta-testing (Cao et al., 2020); however, the linear-evaluation-based approach do not need to take this

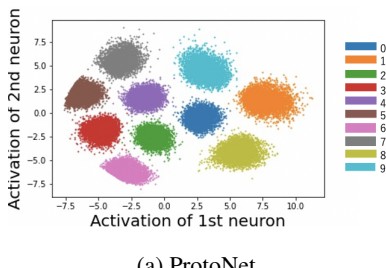 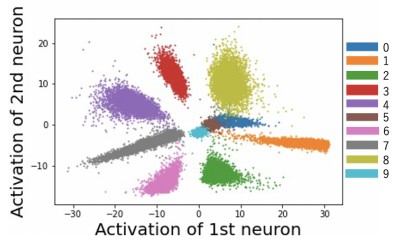

(a) ProtoNet  (b) cross-entropy loss with linear projection layer

Figure 1: Distribution of features extracted using a neural network with two dimensional final layer trained on CIFAR-10 with (a): ProtoNet loss (b): cross-entropy loss with linear projection layer. The ProtoNet features distribute closer to its class center than the features extracted using the model trained on cross-entropy loss with linear projection layer.

into account. However, the linear-evaluation-based approach requires retraining a linear classifier every time a new class appears.

In contrast, a prototype classifier can be applied to any trained feature extractor and does not require model learning in the testing phase. Therefore, a prototype classifier can be a a practical and useful first step for few-shot learning problems. In order to avoid meta-learning during the training phase and the linear evaluation during the testing phase, we focus on using a prototype classifier on the testing phase and training models in a standard classification manner. As we discuss in section 4, we found that when we directly constructed prototypes from the feature vectors extracted using pre-trained models and applied the nearest neighbor method as in the testing phase in ProtoNet , this does not perform as well as the linear-evaluation-based approach.

We hypothesize that the reason is the difference between the loss function in ProtoNet and pre-trained models. As described in section 3, if we consider a prototype as a pseudo sample average of the features in each class, the loss function of ProtoNet can be considered having a regularizing effect that makes it closer to the sample average of the class. Since standard classification training computes cross-entropy loss with additional linear projection to make the features linearly separable, the loss function does not have such an effect and can cause large within-class variance. Figure 1 shows a scatter plot of the features extracted using a neural network with two dimension output trained on cifar-10 with ProtoNet(1a) and cross-entropy loss with a linear projection layer(1b). This figure implies that the features extracted using a model trained in a standard classification manner distributes away from the origin and causes large within-class variance along the direction of the norm of class mean vectors, while that of ProtoNet is more centered to its class means. This phenomenon is also observed in face recognition literatures (Wen et al., 2016; Liu et al., 2017; Wang et al., 2018; Deng et al., 2019).

We now focus on the theoretical analysis on a prototype classifier. A current study (Cao et al., 2020) analyzed an upper bound of risk by using a prototype classifier. The bound depends on the number of shots of a support set, between-class variance, and within-class variance. However, the bound requires the class-conditioned distribution of features to be Gaussian and to have the same covariance matrix among classes. In addition, since the bound does not depend on the norm of the feature vectors, it is not clear from the bound what feature-transformation method can lead to performance improvement. Thus, we derive a novel bound for a prototype classifier.

Our contributions are threefold.

1. We relax the assumption; specifically, the bound does not require that the features distribute on any specific distribution, and each covariance matrix does not have to be the same among classes.

2. We clarify the effect of the variance of the norm of the feature vectors on the performance of a prototype classifier.

3. We investigate the effectiveness of reducing the variance of the norm empirically.

## 2 RELATED WORK

We summarize related work by describing a prototype classifier with meta-learning, linear-evaluation-based approach without meta-learning, and theoretical analysis related to the few-shot learning problem.

**A prototype classifier with meta-learning**    On the basis of the hypothesis that features well distinguished in the training phase are also useful for classifying new classes, constructing one or multiple prototypes for classifying unseen examples is a widely used approach (Vinyals et al., 2016; Snell et al., 2017; Pahde et al., 2021; Ji et al., 2021; Sung et al., 2018; Allen et al., 2019; Doersch et al., 2020; Qi et al., 2018). Certain algorithms compute similarities between multiple prototypes and unseen examples by using their own modules, such as attention mechanism (Vinyals et al., 2016; Doersch et al., 2020), relation network (Sung et al., 2018), reweighting mechanisms by taking into account between-class and within-class interaction (Ji et al., 2021), and latent clusters (Allen et al., 2019). Prototypes are also constructed in a multi-modal way (Pahde et al., 2021). Another line of research is transforming the space of extracted features to a better distinguishable space (Simon et al., 2020; Yoon et al., 2019; Das et al., 2020; Das & Lee, 2020) or taking variance of fetures into account (Bateni et al., 2020) . Because of its convenience, the approach with a prototype classifier is adapted to other domain such as semantic segmentation (Dong & Xing, 2018), text classification (Sun et al., 2019), and speech recognition (Wang et al., 2019a).

**Update-based meta-learning**    In contrast to the approach with a prototype classifier and meta-learning , in update-based meta-learning approaches, model parameters are adjusted in the test phase so that a model can adapt to new classes. Model-agnostic meta-Learning (MAML) and its variants (Finn et al., 2017; 2018; Rajeswaran et al., 2017) search for a good initialization parameters that adapt to new classes with a few labeled data and a few update steps of the parameters. Another approach involves learning an effective update rule for the parameters of a base-learner model through a sequence of training episodes (Bertinetto et al., 2019; Lee et al., 2019). Both approaches require additional learning of hyper-parameters and training time; thus, they prevent quick adaptation to new classes.

**linear-evaluation-based approach without meta-learning**    Interestingly, recent studies have shown that training a new linear classifier with features extracted using a model trained with cross-entropy loss on base-dataset performs comparably with meta-learning based methods (Chen et al., 2019) (Wang et al., 2019b). More effective method for training a new classifier in few-shot settings has been proposed (Yang et al., 2021; Phoo & Hariharan, 2021), such as calibrating distribution generated by a support set (Yang et al., 2021) , self-supervised learning on query data (Phoo & Hariharan, 2021), and distilling knowledge to obtain better embeddings (Tian et al., 2020). Liu et al. (2020) focuses on pretraining phase and found the negative margin in cross-entropy helps improving the performance in few-shot settings. However, similar to update-based meta-learning, these studies require additional hyper-parameters and training to apply to new classes; thus if we tackle these points, we would have an alternative method that is easier and more convenient to use.

**Theoretical analysis of few-shot learning**    Even though much improvement has been empirically made in few-shot learning, theoretical analysis is scarce. In the context of meta-learning, Du et al. (2021) provided a risk bound on the meta-testing phase that is related to the number of meta-training data and meta-testing data. Lucas et al. (2021) derived information-theoretic lower-bounds on mini-max rates of convergence for algorithms that are trained on data from multiple sources and tested on novel data. Cao et al. (2020) derived a new bound on a prototype classifier and theoretically demonstrated that the mismatch regarding the number of shots in a support set between meta-training and meta-testing degrades the performance of prototypical networks, which has been only experimentally observed. However, their bound depends on several assumptions: the class-conditional distributions of features are Gaussian and have the same covariance matrix among classes. In contrast, we derive a novel bound that does not depend on any specific distribution.

# 3 THEORETICAL ANALYSIS OF PROTOTYPE CLASSIFIER IN TERMS OF VARIANCE OF NORM OF FEATURE VECTORS

In this section, we first formulate our problem settings and point out the drawbacks of the current theoretical analysis on a prototype classifier. Next we provide our novel bound of a prototype classifier with the bound related to the variance of the norm of the feature vectors. Finally, we list several methods that can improve the performance of a prototype classifier based on our bound.

## 3.1 PROBLEM SETTING

Let $\mathcal{Y}$ be a space of a class, $\tau$ a probability distribution over $\mathcal{Y}$, $\mathcal{X}$ a space of input data, $\mathcal{D}$ a probability distribution over $\mathcal{X}$, $\mathcal{D}_y$ a probability distribution over $\mathcal{X}$ given a class $y$. We define $\mathcal{D}^{\otimes nk}$ and $\mathcal{D}_y^{\otimes n}$ by $\mathcal{D}_y^{\otimes n} = \Pi_{i=1}^n \mathcal{D}_y$ and $\mathcal{D}^{\otimes nk} \equiv \Pi_{i=1}^n \mathcal{D}_i^{\otimes k}$, respectively. We sample $N$ classes from $\tau$ to form the $N$-way classification problem. Denote by $K$ a number of annotated data in each class and $x \in \mathcal{X}, y \in \mathcal{Y}$ as input data and its class respectively. We define a set of support data of class $c$ sampled from $\tau$ as $S_c = \{\boldsymbol{x}_i \mid (\boldsymbol{x}_i, y_i) \in \mathcal{X} \times \mathcal{Y} \cap y_i = c\}_{i=1}^K$ and a set of support data in the $N$-way $K$-shot classification problem as $S = \bigcup_{c=1}^N S_c$. Suppose a feature extractor computes a function $\phi : \mathcal{X} \to \mathbb{R}^D$, where $D$ is the number of the embedding dimensions. $\overline{\phi(S_c)}$ is defined by $\overline{\phi(S_c)} = \frac{1}{K} \sum_{\boldsymbol{x} \in S_c} \phi(\boldsymbol{x})$. Let $\Phi$ be a space of the extractor function $\phi$. Denote by $\mathcal{M} : \Phi \times \mathcal{X} \times (\mathcal{X} \times \mathcal{Y})^{NK} \to \mathbb{R}^N$ a prototype classifier function that computes probabilities of input $x$ belonging to class $c$ as follows.

$$\mathcal{M}(\phi, \boldsymbol{x}, S)_c = p_{\mathcal{M}}(y = c | \boldsymbol{x}, S, \phi) = \frac{\exp\left(-\|\phi(\boldsymbol{x}) - \overline{\phi(S_c)}\|^2\right)}{\sum_{l=1}^N \exp\left(-\|\phi(\boldsymbol{x}) - \overline{\phi(S_l)}\|^2\right)}, \tag{1}$$

where $\|v\|^2 = \sum_{d=1}^D \left(v^{(d)}\right)^2$, and $v^{(d)}$ is the $d$-th dimension of vector $v$. The prediction of an input $x$, denoted by $\hat{y} \in \mathcal{Y}$, is computed by taking argmax for $\mathcal{M}(\phi, \boldsymbol{x}, S)$, i.e., $\hat{y} = \operatorname{argmax} \mathcal{M}(\phi, \boldsymbol{x}, S)$. We denote by $\mathbb{E}_{z \sim q(z)}[g(z)]$ an operation to take the expectation of $g(z)$ over $z$ distributed as $f(z)$ and we simply denote $\mathbb{E}_{z \sim q(z)}[g(z)]$ as $\mathbb{E}_z[g(z)]$ when $z$ is obviously distributed on $q(z)$. We define $\operatorname{Var}_{z \sim q(z)}[g(z)]$ as an operation to take variance of $g(z)$ over $z$ distributed as $q(z)$. With $\mathbb{I}$ denoting the indicator function, we define expected risk $\mathrm{R}_{\mathcal{M}}$ of a prototype classifier as

$$\mathrm{R}_{\mathcal{M}}(\phi) = \mathbb{E}_{S \sim \mathcal{D}^{\otimes nk}} \mathbb{E}_{c \sim \tau} \mathbb{E}_{\boldsymbol{x} \sim \mathcal{D}_c}[\mathbb{I}[\operatorname{argmax} \mathcal{M}(\phi, \boldsymbol{x}, S) \neq c]. \tag{2}$$

For simplicity, we now discuss the binary classification setting. We show a case of multi-class classification in Appendix A.5 due to lack of space.

Let $c_1$ and $c_2$ denote any pair of classes sampled from $\tau$. We consider that a query data $x$ belongs to class $c_1$ and support sets $S$ consist of class $c_1$'s support set and $c_2$'s support set. Then, equation 2 is written as follows.

$$\mathrm{R}_{\mathcal{M}}(\phi) = \mathbb{E}_{S \sim \mathcal{D}^{\otimes 2k}} \mathbb{E}_{c_1 \sim \tau} \mathbb{E}_{\boldsymbol{x} \sim \mathcal{D}_{c_1}}[\mathbb{I}[\operatorname{argmax} \mathcal{M}(\phi, \boldsymbol{x}, S_{c_1} \cup S_{c_2}) \neq c_1]]. \tag{3}$$

## 3.2 WHAT FEATURE-TRANSFORMATION METHOD EXPECTED TO BE EFFECTIVE?

The current theoretical analysis for a prototype classifier (Cao et al., 2020) has the following two drawbacks (see Appendix A.1 for the details). The first is that the modeling assumption requires a class-conditioned distribution of the features to follow a Gaussian distribution with the same covariance matrix among classes. For example, when we use the ReLU activation function in last layer, it is not normally distributed and the class-conditioned distribution does not have the same covariance matrix as shown in Figure 1. The second drawback is that it is not clear what feature-transformation method can reduce the upper bound. A feature-transformation method to maximize the between-class variance and minimize the within-class variance such as linear discriminant analysis (LDA) (Fukunaga, 1990) and Embedding Space Transformation (EST) (Cao et al., 2020) can be expected to improve performance; however, it is not clear how the second term of the denominator changes.

From Figure 1 , the distribution of each class feature stretches in the direction of the norm of its class-mean feature vector. This property is also observed in metric learning literatures (Wen et al.,

2016; Liu et al., 2017). A model trained in the cross-entropy loss after linear projection from the embedding space to the class-probability space computes a probability of input $x$ belonging to class $c$ given by

$$p(y = c|\boldsymbol{x}, W, \phi) = \frac{\exp\left(\phi(\boldsymbol{x})^\top W_c\right)}{\sum_{j=1}^N \exp\left(\phi(\boldsymbol{x})^\top W_j\right)}, \tag{4}$$

where $W$ is a weight matrix that transforms features from the embedding space to the class-probability space.

Comparing equation 4 with equation 1, we found that equation 4 does not have a regularization term similar to the one appearing in equation 1 that forces the features to be close to its class-mean feature vector. This implies the features extracted using a model trained with the cross-entropy loss on equation 4 are less close to its class-mean feature vector than the ProtoNet loss on equation 1. Through this observation and the property mentioned above, we hypothesis that normalizing the norm of the feature vectors can push the features to each class-mean feature vector and can boost the performance of a prototype classifier trained with cross-entropy loss on equation 4.

### 3.3 RELATING VARIANCE OF NORM TO UPPER BOUND OF EXPECTED RISK

To understand what effect the variance of the norm of the feature vectors has on the performance of a prototype classifier, we analyze how variance contributes to the expected risk when an embedding function $\phi$ is fixed. The following theorem provides a generalization bound for the expected risk of a prototype classifier in terms of the variance of the norm of the feature vectors computed using a feature extractor.

**Theorem 1.** *Let $\mathcal{M}$ be an operation of a prototype classifier on binary classification defined by equation 1. For $\mu_c = \mathbb{E}_{\boldsymbol{x} \sim \mathcal{D}_c}[\phi(\boldsymbol{x})]$, $\Sigma_c = \mathbb{E}_{\boldsymbol{x} \sim \mathcal{D}_c}[(\phi(\boldsymbol{x}) - \mu_c)(\phi(\boldsymbol{x}) - \mu_c)^\top]$, $\mu = \mathbb{E}_{c \sim \tau}[\mu_c]$, $\Sigma = \mathbb{E}_{c \sim \tau}[(\mu_c - \mu)(\mu_c - \mu)^\top]$, and $\mathbb{E}_{c \sim \tau}[\Sigma_c] = \Sigma_\tau$, if $\phi(\boldsymbol{x})$ has the variance of its norm, then the miss-classification risk of the prototype classifier on binary classification $\mathrm{R}_\mathcal{M}$ satisfies*

$$\mathrm{R}_\mathcal{M}(\phi) \leq 1 - \frac{4(\mathrm{Tr}(\Sigma))^2}{\mathbb{E}\mathrm{V}[h_{L2}(\phi(\boldsymbol{x}))] + \mathrm{V}_{\mathrm{Tr}}(\Sigma_{c_1}) + \mathrm{V}_{\mathrm{wit}}(\Sigma_\tau, \Sigma, \boldsymbol{\mu}) + \mathbb{E}\,\mathrm{dist}_{\mathrm{L2}}^2(\boldsymbol{\mu}_{c_1}, \boldsymbol{\mu}_{c_2})}, \tag{5}$$

*where*

$$\mathbb{E}\mathrm{V}[h_{L2}(\phi(\boldsymbol{x}))] = \frac{4}{K} \mathbb{E}_{c \sim \tau}\left[\mathrm{Var}_{\boldsymbol{x} \sim \mathcal{D}_c}\left[\|\phi(\boldsymbol{x})\|^2\right]\right], \tag{6}$$

$$\mathrm{V}_{\mathrm{Tr}}(\Sigma_{c_i}) = \left(\frac{4}{K} + \frac{2}{K^2}\right)\mathrm{Var}_{c \sim \tau}\left[\mathrm{Tr}\left(\Sigma_{c_i}\right)\right], \tag{7}$$

$$\mathrm{V}_{\mathrm{wit}}(\Sigma_\tau, \Sigma, \boldsymbol{\mu}) = \frac{8}{K}\mathrm{Tr}(\Sigma_\tau)\left(\mathrm{Tr}(\Sigma_\tau) + \mathrm{Tr}(\Sigma) + \boldsymbol{\mu}^\top\boldsymbol{\mu}\right) + 4(\mathrm{Tr}(\Sigma) + \boldsymbol{\mu}^\top\boldsymbol{\mu})^2, \tag{8}$$

$$\mathbb{E}\,\mathrm{dist}_{\mathrm{L2}}^2(\boldsymbol{\mu}_{c_1}, \boldsymbol{\mu}_{c_2}) = \mathbb{E}_{c_1, c_2}\left[\left((\boldsymbol{\mu}_{c_1} - \boldsymbol{\mu}_{c_2})^\top(\boldsymbol{\mu}_{c_1} - \boldsymbol{\mu}_{c_2})\right)^2\right]. \tag{9}$$

**Remark.** *The term $\mathbb{E}\mathrm{V}[h_{L2}(\phi(S_y))]$ is the variance of the norm of the feature vectors. The term $\mathrm{V}_{\mathrm{Tr}}(\Sigma_{c_1})$ is the variance of the summation with diagonal element of the covariance matrix from each class embedding; it can be interpreted as the difference of the distribution shape constructed from each class embedding. The term $\mathrm{V}_{\mathrm{wit}}(\Sigma_\tau, \Sigma, \boldsymbol{\mu})$ is correlated with the within-class variance because $\frac{\mathrm{V}_{\mathrm{wit}}(\Sigma_\tau, \Sigma, \boldsymbol{\mu})}{\mathrm{Tr}(\Sigma)}$ is a secondary expression for the ratio of between-class variance and within-class variance. The term $\mathbb{E}\,\mathrm{dist}_{\mathrm{L2}}^2(\boldsymbol{\mu}_{c_1}, \boldsymbol{\mu}_{c_2})$ is the expectation of the Euclidean distance between the class mean vectors .*

This bound has the following properties.

1. Its derivation does not require the features to be distributed on any specific distributions and the class-conditioned covariance matrix does not have to be the same among classes

2. The bound can decrease when any of the following statistics decreases with fixed between-class variance ($\Sigma$): (i) the variance of the norm of the feature vectors, as what we discussed in Section 3.2, (ii) the difference in the distribution shape constructed from each class embedding, (iii) the within-class variance ($\Sigma_\tau$), (iv) the Euclidean distance between the class-mean vectors.

As a result, our bound loosens the modeling assumption of Theorem2 in appendix A.1 and has its property in 2-(iii). We show the proof in Appendix A.2.

### 3.4 FEATURE-TRANSFORMATION METHODS

We hypothesize from Theorem 1 that in addition to a feature-transformation method related to equation 6, eliminating the affect from the difference in the distribution shape among classes and lowering the ratio of between-class variance to within-class variance can improve the performance of a prototype classifier. Regarding equation 9, the ratio of the Euclidean distance between the class mean vectors and the between-class variance is supposed to be constant. Thus, we focus on the transformation methods relating to equation 6, equation 7 and equation 8. We analyze the following feature-transformation methods: $L_2$-normalization ($L_2$-norm), variance-normalization, LDA, EST, and EST+$L_2$-norm. We show the detail of each method in Section A.3 due to lack of space.

## 4 EXPERIMENTAL EVALUATION

In this section, we experimentally analyzed the effectiveness of the feature-transformation methods mentioned in Section3.4. The center loss (Wen et al., 2016) and affinity loss (Hayat et al., 2019) have been proposed to efficiently pull the features of the same class to their centers in the training phase; however, we focus on a widely used pre-trained model in this experiments following the line of studies Chen et al. (2019) and Tian et al. (2020).

### 4.1 DATASETS AND EVALUATION PROTOCOL

**miniImageNet.** The *mini*ImageNet dataset (Vinyals et al., 2016) is a standard bench-mark for few-shot learning algorithms for recent studies. It contains 100 classes randomly sampled from ImageNet. Each class contains 600 images. We follow the widely-used splitting protocol (Ravi & Larochelle, 2017) to split the dataset into 64/16/20 for training/validation/testing respectively.

**tieredImageNet** The *tiered*ImageNet dataset (Ren et al., 2018) is another subset of ImageNet but has 608 classes. These classes are grouped into 34 higher level categories in accordance with the ImageNet hierarchy and these categories are split into 20 training (351 classes), 6 validation (97 classes), 8 testing categories (160 classes). This splitting protocol ensures that the training set is distinctive enough from the testing set and makes the problem more realistic since we generally cannot assume that test classes will be similar to those seen in training.

**CIFAR-FS** The CIFAR-FS dataset (Bertinetto et al., 2019) is a recently proposed fewshot image classification benchmark, consisting of all 100 classes from CIFAR-100 (Krizhevsky et al.). The classes are randomly split into 64, 16, and 20 for meta-training, meta-validation, and meta-testing respectively. Each class contains 600 images of size $32 \times 32$.

**FC100** The FC100 dataset (Lee et al., 2019) is another dataset derived from CIFAR-100 (Krizhevsky et al.), containing 100 classes which are grouped into 20 categories. These categories are split into 12 categories for training (60 classes), from 4 categories for validation (20 classes), 4 categories for testing (20 classes).

### 4.2 IMPLEMENTATION DETAILS

We compared the feature-transformation methods applied on against ProtoNet (Snell et al., 2017), Baseline and Baseline++ (Chen et al., 2019). In Baseline, the linear projection layer is trained on a support set and in Baseline++, the norm of the linear projection layer and features are normalized to be constant. We call Baseline and Baseline++ as "linear evlauation" methods. We also compared with the feature-transformation method proposed in a previous study (Wang et al., 2019b). In that study they transformed the features so that the mean of all features to be origin before $L_2$-normalization and then use a prototype classifier without training a new linear classifier. We call this operation as centering+$L_2$-norm. We re-implemented these methods following the training procedure in previous study (Chen et al., 2019). In the pre-training stage, where the cross-entropy loss was used and meta-learning was not used, we trained 400 epochs with a batch size of 16. In the meta-training stage for ProtoNet, we trained $60,000$ episodes for 1-shot and $40,000$ episodes

Table 1: Classification accuracies on *mini*ImageNet and *tiered*ImageNet of ProtoNet , linear evaluation methods (Chen et al., 2019), and ours. The best performing methods and any other runs within $95\%$ confidence margin are in bold.

| Method | *mini*ImageNet | | | | *tiered*ImageNet | | | |
|---|---|---|---|---|---|---|---|---|
| | ResNet12 | | ResNet18 | | ResNet12 | | ResNet18 | |
| | 1-shot | 5-shot | 1-shot | 5-shot | 1-shot | 5-shot | 1-shot | 5-shot |
| ProtoNet | 53.48% | 73.56% | 56.26% | 74.02% | 55.40% | 77.67% | 60.50% | 81.40% |
| B@FT | 54.54% | **76.50%** | 55.41% | **76.95%** | 61.67% | **81.62%** | 63.38% | **83.18%** |
| B+@FT | 56.33% | 74.62% | 55.07% | 74.15% | 63.02% | 81.07% | 64.20% | 81.62% |
| B@CL2 | **58.66%** | 75.98% | 57.67% | 70.78% | **64.88%** | 80.42% | **65.26%** | 81.63% |
| B+@CL2 | 57.50% | 74.00% | 57.00% | 74.06% | 63.31% | 79.19% | 65.67% | 81.41% |
| B | 46.36% | 73.97% | 43.86% | 72.36% | 50.60% | 78.10% | 56.16% | 80.33% |
| B@L2-N | 57.18% | **77.12%** | 56.57% | **76.44%** | **64.04%** | **81.73%** | **65.19%** | **82.93%** |
| B@V-N | – | 63.55% | – | 62.78% | – | 66.08% | – | 75.56% |
| B@LDA | – | 73.75% | – | 73.54% | – | 77.44% | – | 80.85% |
| B@EST | 51.28% | 72.80% | 44.19% | 72.99% | 53.90% | 78.09% | 57.12% | 80.59% |
| B@EST+L2-N | **58.00%** | **76.90%** | 56.39% | **76.24%** | **64.54%** | **81.40%** | 64.71% | **83.24%** |
| B+ | 41.18% | 73.97% | 36.80% | 63.76% | 46.52% | 73.85% | 48.27% | 75.87% |
| B+@L2-N | **57.96%** | 75.38% | **57.21%** | 74.89% | **64.96%** | **81.08%** | **65.40%** | 82.49% |
| B+@V-N | – | 55.92% | – | 54.31% | – | 62.19% | – | 64.33% |
| B+@LDA | – | 69.60% | – | 74.38% | – | 72.88% | – | 75.55% |
| B+@EST | 47.11% | 69.01% | 47.21% | 69.11% | 52.01% | 74.26% | 53.49% | 75.38% |
| B+@EST+L2-N | **58.32%** | **77.49%** | **57.00%** | **77.13%** | 57.63% | **80.99%** | 60.87% | 81.80% |

for 5-shot tasks. We used the validation set to select the training episodes with the best accuracy. In each episode, we sampled $N$ classes to form $N$-way classification (In meta-training $N$=20 and meta-testing $N$=5 following the original study of ProtoNet (Snell et al., 2017)). For each class, we selected $K$ labeled instances as our support set and 16 instances for the query set for a $K$-shot task.

In the linear evaluation or meta-testing stage for all methods, we averaged the results over 600 trials. In each trial, we randomly sampled 5 classes from novel classes, and in each class, we also selected $K$ instances for the support set and 16 for the query set. For Baseline and Baseline++, we used the entire support set to train a new classifier for 100 iterations with a batch size of 4. With ProtoNet, we used the models trained in the same shots as meta-testing stage since the mismatch of number of shots between meta-training and meta-testing degrades performance (Cao et al., 2020). All methods were trained from scratch, and the Adam optimizer with initial learning rate $10^{-3}$ was used. We applied standard data augmentation including random crop, horizontal-flip, and color jitter in both the training stages. We used a ResNet12 network and ResNet18 network following the previous study (Chen et al., 2019; Yoon et al., 2019).

Since LDA constructs a within-class covariance matrix from a support set and variance-normalization computes sample variance, we did not investigate the LDA score and variance-normalization score of 1-shot settings. For LDA we set $\lambda = 0.0001$ for equation 18 in Appendix A.3 and for EST we set the dimensions of the projected space to 60 following the settings of the original study (Cao et al., 2020).

### 4.3 RESULTS

We present the experimental results on Table 1 and Table 2 on the basis of backbones with ResNet12 and ResNet18 for a comprehensive comparison. Table 1 shows the results on *mini*ImageNet and *tiered*ImageNet, and Table 2 shows the results on CIFARFS and FC100. We show the detailed performance results with $95\%$ confidence margin and other transformation methods in A.6.

**Method notations in experimental results** We denote B@FT and B+@FT as the linear evaluation methods: Baseline and Baseline++. We also denote B@CL2, B+@CL2 as centering+$L_2$-norm (Wang et al., 2019b) on features trained with Baseline and Baseilne++. B, B@L2-N, B@V-N, B@LDA, B@EST, and B@EST+L2-N stands for a prototype classifier with applying no transformation, $L_2$-norm, variance-normalization, LDA, EST, and EST+L2-N on the features extracted using a model pre-trained in the same manner as B@FT. B+, B+@L2-N, B+@V-N, B+@LDA, B+@EST, and B+@EST+L2-N stands for a prototype classifier with applying no transformation,

Table 2: Classification accuracies on CIFARFS and FC100 of ProtoNet, linear evaluation methods (Chen et al., 2019), and ours. The best performing methods and any other runs within $95\%$ confidence margin are in bold.

| Method | CIFARFS | | | | FC100 | | | |
| --- | --- | --- | --- | --- | --- | --- | --- | --- |
| | ResNet12 | | ResNet18 | | ResNet12 | | ResNet18 | |
| | 1-shot | 5-shot | 1-shot | 5-shot | 1-shot | 5-shot | 1-shot | 5-shot |
| ProtoNet | 58.65% | 75.33% | 62.05% | **78.25%** | 35.56% | 51.12% | 36.02% | 51.02% |
| B@FT | 55.97% | **76.50%** | 56.46% | 77.43% | 39.72% | **56.04%** | 40.06% | 57.04% |
| B+@FT | **61.08%** | 76.15% | 61.34% | **77.86%** | 36.01% | 50.73% | 36.93% | 50.41% |
| B@CL2 | 58.61% | 74.73% | 58.09% | 76.52% | 41.58% | 55.82% | 41.51% | 56.44% |
| B+@CL2 | **61.59%** | 76.10% | **63.17%** | **77.49%** | 37.51% | 50.38% | 38.30% | 51.06% |
| B | 44.12% | 72.47% | 47.86% | 75.66% | 31.60% | 52.50% | 35.90% | 55.20% |
| B@L2-N | 59.43% | **77.45%** | 58.51% | 77.43% | 40.34% | **56.61%** | 40.49% | 57.71% |
| B@V-N | – | 44.66% | – | 67.72% | – | 41.99% | – | 49.69% |
| B@LDA | – | 75.73% | – | 75.25% | – | 52.90% | – | 55.44% |
| B@EST | 52.62% | 72.80% | 55.57% | 75.10% | 44.45% | 52.97% | 47.49% | 55.68% |
| B@EST+L2-N | **60.70%** | **77.04%** | 61.14% | **77.48%** | **47.57%** | **56.96%** | **50.13%** | **59.94%** |
| B+ | 45.73% | 73.97% | 36.80% | 63.76% | 29.72% | 46.85% | 30.76% | 47.62% |
| B+@L2-N | **61.07%** | 76.71% | **63.48%** | **77.86%** | 37.16% | 51.10% | 38.55% | 50.15% |
| B+@V-N | – | 57.20% | – | 57.16% | – | 36.76% | – | 40.66% |
| B+@LDA | – | 68.58% | – | 69.86% | – | 47.10% | – | 47.63% |
| B+@EST | 50.18% | 70.32% | 49.82% | 71.07% | 37.60% | 46.81% | 39.92% | 47.65% |
| B+@EST+L2-N | 59.83% | 76.32% | **63.00%** | **77.99%** | 40.88% | 50.45% | 41.65% | 50.74% |

$L_2$-norm, and variance-normalization on the features extracted using the same pre-trained model as with B+@FT.

**Comparison of the feature transformation methods with ProtoNet, linear evaluation methods, and centering+$L_2$-norm** From Table 1 and Table 2, we can observe that the prototype classifier with $L_2$-norm and EST+$L_2$-norm performs comparably with ProtoNet and the linear-evaluation-based approach in all settings and EST+$L_2$-norm performs the best among the data transformation methods. Comparing the feature-transformation methods described in Section 3.4 with centering+$L_2$-norm (Wang et al., 2019b), centering+$L_2$-norm can slightly improve the performance of the prototype classifier in several 1-shot settings . However, in 5-shot settings, the boost decreases and even performs worse than $L_2$-norm, e.g. *mini*ImageNet and *tiered*ImageNet with ResNet12.

**Comparison of the feature transformation methods with B and B+** In the 1-shot setting, although EST falls short of ProtoNet and linear-evaluation-based approach, it also improves the performance of a prototype classifier. The performance gain of both $L_2$-norm, EST and EST+$L_2$-norm decrease when the number of shots increases. This phenomenon can be explained through Theorem 1. The term relating to the variance of the norm and the ratio of the between-class variance to the within-class variance depends on $K$.Since the term diminishes with increasing $K$, the performance gain of the feature-transformation methods decreases.

**Discussion on the feature transformation methods** Surprisingly, variance normalization performs worse than applying no feature transformation. This can be explained by the following two reasons. The first is that the term in the bound related to the difference among the class distribution shapes is smaller than other terms such as the variance of the norm and the ratio of the between-class variance to the within-class variance. Figure 2 shows the term associated with the bound computed from each dataset and backbone. The larger the value is, the more it contributes to larger risk. We found the term associated with equation 7 is small compared with other values and thus variance-normalization did not worked well in our experiment. The second reason is that estimating variance with a support set is unstable and make prediction unreliable in few-shot learning. LDA and EST in 5-shot setting does not improve performance so much compared with $L_2$-norm variants while in the 1-shot settings, EST helps improving performance. This is because in the 5-shot settings, the values computed from equation 8 gets smaller with larger $K$ and the effect of equation 8 on the risk of a prototype classifier in the 1-shot settings is larger. Especially, EST outperforms $L_2$-norm in FC100 with the 1-shot setting. We found from Figure 2 the features of FC100 shows the largest ratio of the

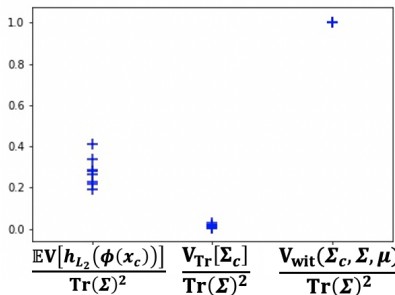 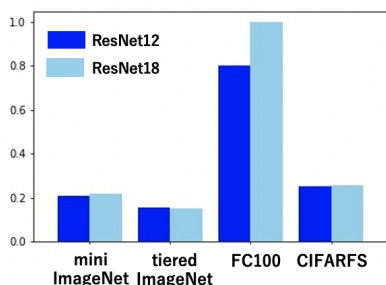

Figure 2: Left: We plotted the values shown in equations 6,7,8 divided by $\mathrm{Tr}(\Sigma)^2$. The values are computed from test-split of each dataset with ResNet12 and ResNet18. We scaled the values so that each dataset's $\frac{V_{\mathrm{wit}}(\Sigma_\tau, \Sigma, \mu)}{\mathrm{Tr}(\Sigma)^2}$ to be 1 for simplicity. Right: We showed the ratio of the within-class variance to the between-class variance computed from the test-split's features extracted using each backbone trained on each corresponding dataset. We scaled the values so that the maximum value to be 1 for simplicity.

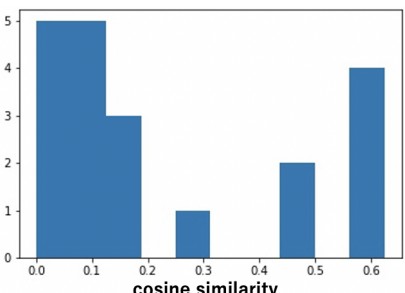 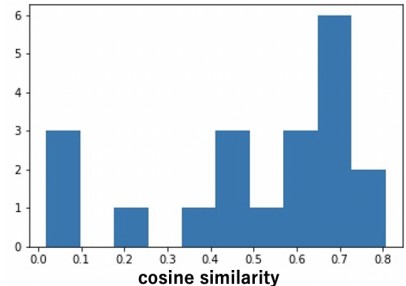

Figure 3: Histogram of cosine similarities with class mean vectors and largest eigen vectors of Left:$\Sigma$, Right:$\Sigma_c$

within-class to between-class variance among all dataset. Thus the method reducing the ratio works better with FC100 features than any other dataset's features.

## 4.4 EFFECT ON COVARIANCE MATRIX

We further analyzed how $\mathrm{Tr}(\Sigma)$ and the trace of class $c$'s covariance matrix $\mathrm{Tr}(\Sigma_c)$ fluctuate with $L_2$-norm. Since $L_2$-norm lowers the variance in the direction of the normalized vector, if the covariance matrix stretches in the direction of the class mean vector, $L_2$-norm can reduce its trace. Thus we analyzed the cosine similarities between class-mean feature vectors and the eigenvectors with the largest eigen value of the covariance matrices. Figure 3 illustrates the distribution of cosine similarities of each class-mean feature vector and the eigenvectors in test class of *mini*ImageNet. This figure shows $\Sigma_c$ tends to stretches to the direction of class mean vector and $\Sigma$ stretches in the orthogonal direction to each class-mean feature vector. From this observation, we found that $L_2$-norm can reduce $\mathrm{Tr}(\Sigma_c)$ and has less effect on $\mathrm{Tr}(\Sigma)$. $L_2$-norm can reduce the variance of the norm and the ratio of the within-class variance to the between-class variance.

## 5 CONCLUSION

We theoretically and experimentally analyzed how the variance of the norm of feature vectors affects the performance of a prototype classifier and found that using EST+$L_2$-norm makes the classifier comparable with ProtoNet and the linear-evaluation-based approach. We also found that when the number of shots in a support set increases, the performance gain from a feature-transformation method decreases, which is consistent with the results of the theoretical analysis. A prototypical classifier is expected to be a practically useful first step in tackling few-shot learning problems because of its simplicity.

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

# A    APPENDIX

## A.1    EXISTING UPPER BOUND ON EXPECTED RISK FOR PROTOTYPE CLASSIFIER

To analyze the behavior of a prototype classifier, we start from the current study (Cao et al., 2020). The following theorem is the upper bound of the expected risk of prototypical networks with the next conditions.

- The probability distribution of an extracted feature $\phi(\boldsymbol{x})$ given its class $y = c$ is Gaussian i.e $\mathcal{D}_y = \mathcal{N}(\mu_c, \Sigma_c)$, where $\mu_c = \mathbb{E}_{\boldsymbol{x} \sim \mathcal{D}_c}[\phi(\boldsymbol{x})]$ and $\Sigma_c = \mathbb{E}_{\boldsymbol{x} \sim \mathcal{D}_c}[(\phi(\boldsymbol{x}) - \mu_c)(\phi(\boldsymbol{x}) - \mu_c)^\top]$.
- All class-conditioned distributions have the same covariance matrix, i.e., $\forall (c, c'), \Sigma_c = \Sigma_{c'}$.

**Theorem 2.** *Let $\mathcal{M}$ be an operation of a prototype classifier on binary classification defined by equation 1. Then for $\mu = \mathbb{E}_{c \sim \tau}[\mu_c]$ and $\Sigma = \mathbb{E}_{c \sim \tau}[(\mu_c - \mu)(\mu_c - \mu)^\top]$, the miss-classification risk of the prototype classifier on binary classification $\mathrm{R}_\mathcal{M}$ satisfies*

$$\mathrm{R}_\mathcal{M}(\phi) \leq 1 - \frac{4 \operatorname{Tr}(\Sigma)^2}{8(1 + 1/k)^2 \operatorname{Tr}(\Sigma_c^2) + 16(1 + 1/k) \operatorname{Tr}(\Sigma \Sigma_c) + \mathbb{E} \operatorname{dist}_{\mathrm{L2}}^2(\boldsymbol{\mu}_{c_1}, \boldsymbol{\mu}_{c_2})}, \quad (10)$$

*where $\mathbb{E} \operatorname{dist}_{\mathrm{L2}}^2(\boldsymbol{\mu}_{c_1}, \boldsymbol{\mu}_{c_2}) = \mathbb{E}_{c_1, c_2} \left[ ((\boldsymbol{\mu}_{c_1} - \boldsymbol{\mu}_{c_2}) \top (\boldsymbol{\mu}_{c_1} - \boldsymbol{\mu}_{c_2}))^2 \right].$*

We show the detail of the derivation in Appendix A.2.

## A.2    DERIVATION DETAILS OF THEOREM 2

we will explain the rough sketch of derivation of Theorem 2. In prototype classifier, from equation 1 and equation 3, $R_\mathcal{M}$ is written with sigmoid function $\sigma$ as follows:

$$R_\mathcal{M}(\phi) = \Pr_{c_1, c_2 \sim \tau, \boldsymbol{x} \sim \mathcal{D}_{c_1}, S \sim \mathcal{D} \otimes 2K} \left( \sigma(\|\phi(\boldsymbol{x}) - \overline{\phi(S_{c_2})}\| - \|\phi(\boldsymbol{x}) - \overline{\phi(S_{c_1})}\|) \leq \frac{1}{2} \right)$$
$$= \Pr(\alpha < 0), \quad (11)$$

where $\alpha \triangleq \|\phi(\boldsymbol{x}) - \overline{\phi(S_{c_2})}\| - \|\phi(\boldsymbol{x}) - \overline{\phi(S_{c_1})}\|$. we bound equation 11 with expectation and variance of $\alpha$ by following proposition.

**Proposition 1.** *From the one-sided Chebyshev's inequality, it immediately follows that:*

$$\mathrm{R}_\mathcal{M}(\phi) = \Pr(\alpha < 0) \leq 1 - \frac{\mathbb{E}_{S \sim \mathcal{D} \otimes 2k} \mathbb{E}_{c_1, c_2 \sim \tau} \mathbb{E}_{\boldsymbol{x} \sim \mathcal{D}_{c_1}}[\alpha]^2}{\operatorname{Var}_{S, c_1, c_2, \boldsymbol{x}}[\alpha] + \mathbb{E}_{S \sim \mathcal{D} \otimes 2k} \mathbb{E}_{c_1, c_2 \sim \tau} \mathbb{E}_{\boldsymbol{x} \sim \mathcal{D}_{c_1}}[\alpha]^2}. \quad (12)$$

equation 11 can be further write down as follows by Law of Total Expectation

$$\operatorname{Var}_{S, c, \boldsymbol{x}}(\alpha) = \mathbb{E}_{S \sim \mathcal{D} \otimes 2k} \mathbb{E}_{c_1, c_2 \sim \tau} \mathbb{E}_{\boldsymbol{x} \sim \mathcal{D}_c}[\alpha^2] - \left( \mathbb{E}_{S \sim \mathcal{D} \otimes 2k} \mathbb{E}_{c_1, c_2 \sim \tau} \mathbb{E}_{\boldsymbol{x} \sim \mathcal{D}_{c_1}}[\alpha] \right)^2$$
$$= \mathbb{E}_{c_1, c_2} \mathbb{E}_{\boldsymbol{x}, S}[\alpha^2 | c_1, c_2] - \left( \mathbb{E}_{S \sim \mathcal{D} \otimes 2k} \mathbb{E}_{c_1, c_2 \sim \tau} \mathbb{E}_{\boldsymbol{x} \sim \mathcal{D}_{c_1}}[\alpha] \right)^2$$
$$= \mathbb{E}_{c_1, c_2}[\operatorname{Var}_{\boldsymbol{x}, S}(\alpha | c_1, c_2) + \mathbb{E}_{\boldsymbol{x}, S}[\alpha | c_1, c_2]^2] - \mathbb{E}_{c_1, c_2 \sim \tau} \mathbb{E}_{\boldsymbol{x} \sim \mathcal{D}_{c_1}} \mathbb{E}_{S \sim \mathcal{D} \otimes 2K}[\alpha]^2.$$

Therefore,

$$\Pr(\alpha < 0) \leq 1 - \frac{\mathbb{E}_{S \sim \mathcal{D}^{\otimes 2k}} \mathbb{E}_{c_1, c_2 \sim \tau} \mathbb{E}_{\boldsymbol{x} \sim \mathcal{D}_c}[\alpha]^2}{\mathbb{E}_{c_1, c_2}[\mathrm{Var}_{\boldsymbol{x} \sim D_{c_1}, S}[\alpha] + \mathbb{E}_{\boldsymbol{x} \sim D_{c_1}, S}[\alpha]^2]}. \tag{13}$$

We write down the expection and variance of $\alpha$ with following Lemmas 3 and 4.

**Lemma 3.** *Under the same notation and assumptions as Theorem 2, then,*

$$\mathbb{E}_{S \sim \mathcal{D}^{\otimes 2k}} \mathbb{E}_{\boldsymbol{x} \sim \mathcal{D}_{c_1}}[\alpha] = (\boldsymbol{\mu}_{c_1} - \boldsymbol{\mu}_{c_2})^\top (\boldsymbol{\mu}_{c_1} - \boldsymbol{\mu}_{c_2})$$
$$\mathbb{E}_{c_1, c_2 \sim \tau} \mathbb{E}_{\boldsymbol{x} \sim \mathcal{D}_{c_1}} \mathbb{E}_{S \sim \mathcal{D}^{\otimes 2K}}[\alpha] = 2\mathrm{Tr}(\Sigma).$$

**Lemma 4.** *Under the same notation and assumptions as Theorem 2, then,*

$$\mathbb{E}_{c_1, c_2}[\mathrm{Var}_{\boldsymbol{x}, S}[\alpha | c_1, c_2]] \leq 8 \left(1 + \frac{1}{K}\right) \mathrm{Tr}\left(\Sigma_\tau((1 + \frac{1}{K})\Sigma_\tau + 2\Sigma)\right). \tag{14}$$

The proofs of the above lemmas are in the current study (Cao et al., 2020).

With Proposition 1 and Lemma 3, Lemma 4, we obtain Theorem 2.

## A.3 DETAIL OF FEATURE-TRASNFORMING METHODS

$L_2$ **normalization ($L_2$-norm)**    From equation 6, normalizing the norm of the feature vectors can improve the performance of a prototype classifier. We denote a function normalizing the norm as $\psi_{L2}$ given by

$$\psi_{L2}(\phi(\boldsymbol{x})) = \frac{\phi(\boldsymbol{x})}{\|\phi(\boldsymbol{x})\|}. \tag{15}$$

**variance-normalization**    From equation 7, normalizing the Euclidean distance in equation 1 by the variance of each class embedding can improve the performance of a prototype classifier. This can be interpreted as a variation of the Mahalanobis distance with a diagonal covariance matrix. We denote a function that measures the distance of two vectors normalized by its variance at each dimension as follows:

$$\psi_{\mathrm{maha}}(\phi(\boldsymbol{x})_1), \phi(\boldsymbol{x})_2)) = \sum_{d=1}^{D} \frac{\left(\phi(\boldsymbol{x})_1)^{(d)} - \phi(\boldsymbol{x})_2)^{(d)}\right)^2}{\overline{\mathrm{Var}[\phi(\boldsymbol{x})_1)^{(d)}]}}, \tag{16}$$

where $\overline{\mathrm{Var}[z]}$ is sample variance of $z$. equation 1 is rewritten with $\psi_{\mathrm{maha}}(\phi(\boldsymbol{x}_1), \phi(\boldsymbol{x}_2))$ as the following equation.

$$\mathcal{M}(\phi, \boldsymbol{x}, S)_j = \frac{\exp\left(-\psi_{\mathrm{maha}}\left(\phi(\boldsymbol{x}), \overline{\phi(S_j)}\right)\right)}{\sum_{l=1}^{N} \exp\left(-\psi_{\mathrm{maha}}\left(\phi(\boldsymbol{x}), \overline{\phi(S_l)}\right)\right)}. \tag{17}$$

This normalization is similar to Bateni et al. (2020)'s work but our method differs in that we do not need to do meta-learning and we do not need to add other modules to the backbone network.

**LDA**    From equation 8, increasing the ratio of the between-class variance to the within-class variance can improve the performance a prototype classifier. LDA (Fukunaga, 1990) is widely used to search for a projection space that maximizes the between-class variance and minimizes the within-class variance. It computes the eigenvectors of a matrix $\hat{\Sigma}_\tau^{-1} \hat{\Sigma}$, where $\hat{\Sigma}$ is the covariance matrix of prototypes and $\hat{\Sigma}_\tau$ is the class-conditioned covariance matrix. Since the number of data is small in few-shot settings, $\hat{\Sigma}_\tau^{-1}$ cannot be estimated stably and we add a regularizer term to $\hat{\Sigma}_\tau$ and define it as $\hat{\Sigma}_{\tau \mathrm{reg}}$.

$$\hat{\Sigma}_{\tau \mathrm{reg}} = \hat{\Sigma}_\tau + \lambda I, \tag{18}$$

where $\lambda \in R^1, I \in R^{D \times D}$ is identity matrix.

**EST** Since computation of LDA is unstable, we also analyze the effect of EST (Cao et al., 2020). EST computes eigenvectors of a matrix $\hat{\Sigma} - \rho\hat{\Sigma}_\tau$: the difference between the covariance matrix of the class mean vectors and the class mean covariance matrix with weight parameter $\rho$. Similar to LDA, EST also searches for the projection space that maximizes the $\Sigma$ and minimizes the $\Sigma_\tau$. We compute the eigenvectors of the matrix with two small modification. The first is that we remove hyper-parameter $rho$ and then simply computes the eigenvectors of $\hat{\Sigma} - \hat{\Sigma}_\tau$. The second is that we constructs the matrix from a support set in the multi-shot setting while we follows the original study in the 1-shot setting.

**EST+$L_2$-norm** We hypothesize that the combination of the transforming methods can improve the performance of a prototype classifier independently from each other. Specifically, we focus on reducing equation 6 and equation 8 by combining **EST** and $L_2$-**norm**. We first apply EST to reduce equation 8 and after that we apply $L_2$-norm to reduce equation 6. At the end of the operation we want the variance of the norm to be 0, thus we apply EST and after that we apply $L_2$-norm.

**variance-normalization+$L_2$-norm** We focus on reducing equation 6 and equation 7 by combining **variance-normalization** and $L_2$-**norm**. Following the similar procedure of **EST+$L_2$-norm**, we first apply variance-normalization and after that we apply $L_2$-norm. We show the experimental results of this transformation methods in A.6.

**LDA+$L_2$-norm** We focus on reducing equation 8 and equation 7 by combining **LDA** and $L_2$-**norm**. Following the similar procedure of **EST+$L_2$-norm**, we first apply LDA and after that we apply $L_2$-norm. We show the experimental results of this transformation methods in A.6.

## A.4    DERIVATION DETAILS OF THEOREM 1

We start the proof from equation 13. We first prove the following Lemma 5 related to the expectation statistics of $\alpha$ in equation 13.

**Lemma 5.** *Under the same notations and assumptions as Theorem 1, then,*

$$\mathbb{E}_{S\sim\mathcal{D}^{\otimes 2k}}\mathbb{E}_{\boldsymbol{x}\sim\mathcal{D}_{c_1}}[\alpha] = \frac{1}{K}\left(\mathrm{Tr}(\Sigma_{c_2}) - \mathrm{Tr}(\Sigma_{c_1})\right) + (\boldsymbol{\mu}_{c_1} - \boldsymbol{\mu}_{c_2})\top(\boldsymbol{\mu}_{c_1} - \boldsymbol{\mu}_{c_2})$$

$$\mathbb{E}_{c_1,c_2\sim\tau}\mathbb{E}_{\boldsymbol{x}\sim\mathcal{D}_{c_1}}\mathbb{E}_{S\sim\mathcal{D}_{\otimes 2K}}[\alpha] = 2\mathrm{Tr}(\Sigma).$$

*Proof.* First, from the definition of $\alpha$, we split $\mathbb{E}_{S\sim\mathcal{D}^{\otimes 2k}}\mathbb{E}_{\boldsymbol{x}\sim\mathcal{D}_{c_1}}[\alpha]$ in to two parts and examine them seperately.

$$\mathbb{E}_{S\sim\mathcal{D}^{\otimes 2k}}\mathbb{E}_{\boldsymbol{x}\sim\mathcal{D}_{c_1}}[\alpha] = \underbrace{\mathbb{E}\left[\left\|\phi(\boldsymbol{x}) - \overline{\phi(S_{c_2})}\right\|^2\right]}_{(i)} - \underbrace{\mathbb{E}\left[\left\|\phi(\boldsymbol{x}) - \overline{\phi(S_{c_1})}\right\|^2\right]}_{(ii)}. \tag{19}$$

In regular conditions, for random vector $X$, the expectation of the norm is

$$\mathbb{E}[X^\top X] = \mathrm{Tr}(\mathrm{Var}(X)) + \mathbb{E}[X]^\top\mathbb{E}[X], \tag{20}$$

and the variance of the vector is

$$\mathrm{Var}(X) = \mathbb{E}[XX^\top] - \mathbb{E}[X]E[X]^\top \tag{21}$$

$$\Sigma_{c_i} \triangleq \mathrm{Var}_{\boldsymbol{x}\sim\mathcal{D}_{c_i}}(\phi(\boldsymbol{x})). \tag{22}$$

Hence,

$$(i) = \mathbb{E}_{\boldsymbol{x}\sim D_{c_1}}\mathbb{E}_S\left[\left\|\phi(\boldsymbol{x}) - \overline{\phi(S_{c_2})}\right\|^2\right]$$

$$= \mathrm{Tr}\left(\mathrm{Var}_{\boldsymbol{x}\sim D_{c_1},S}\left[\phi(\boldsymbol{x}) - \overline{\phi(S_{c_2})}\right]\right) + \mathbb{E}_{\boldsymbol{x}}\mathbb{E}_S\left[\phi(\boldsymbol{x}) - \overline{\phi(S_{c_2})}\right]^\top\mathbb{E}_{\boldsymbol{x}}\mathbb{E}_S\left[\phi(\boldsymbol{x}) - \overline{\phi(S_{c_2})}\right], \tag{23}$$

where the first term inside the trace can be expanded as:

$$\mathrm{Var}\left[\phi(\boldsymbol{x}) - \overline{\phi(S_{c_2})}\right] = \mathbb{E}\left[\left(\phi(\boldsymbol{x}) - \overline{\phi(S_{c_2})}\right)\left(\phi(\boldsymbol{x}) - \overline{\phi(S_{c_2})}\right)^\top\right] - (\boldsymbol{\mu}_{c_1} - \boldsymbol{\mu}_{c_2})(\boldsymbol{\mu}_{c_1} - \boldsymbol{\mu}_{c_2})^\top$$

$$= \mathrm{Var}\left[\phi(\boldsymbol{x})\right] + \mathbb{E}\left[\phi(\boldsymbol{x})\right]\mathbb{E}\left[\phi(\boldsymbol{x})\right]^\top + \mathrm{Var}\left[\overline{\phi(S_{c_2})}\right] + \mathbb{E}\left[\overline{\phi(S_{c_2})}\right]\mathbb{E}\left[\overline{\phi(S_{c_2})}\right]^\top$$

$$- \boldsymbol{\mu}_{c_2}\boldsymbol{\mu}_{c_1}^\top - \boldsymbol{\mu}_{c_1}\boldsymbol{\mu}_{c_2}^\top - (\boldsymbol{\mu}_{c_1} - \boldsymbol{\mu}_{c_2})(\boldsymbol{\mu}_{c_1} - \boldsymbol{\mu}_{c_2})^\top$$

$$= \Sigma_{c_1} + \frac{1}{K}\Sigma_{c_2} \quad \text{(Last terms cancel out).} \tag{24}$$

The second term in equation 23 is simply as follows.

$$\mathbb{E}_{\boldsymbol{x}\sim D_{c_1}}\mathbb{E}_S\left[\phi(\boldsymbol{x}) - \overline{\phi(S_{c_2})}\right] = \boldsymbol{\mu}_{c_1} - \boldsymbol{\mu}_{c_2}. \tag{25}$$

From equation 24 and equation 25 we obtain

$$(i) = \mathrm{Tr}(\Sigma_{c_1}) + \frac{1}{K}\mathrm{Tr}(\Sigma_{c_2}) + (\boldsymbol{\mu}_{c_1} - \boldsymbol{\mu}_{c_2})^\top(\boldsymbol{\mu}_{c_1} - \boldsymbol{\mu}_{c_2}). \tag{26}$$

Similarly for ii,

$$(ii) = \mathrm{Tr}\left(\mathrm{Var}_{\boldsymbol{x}\sim D_{c_1},S}\left[\phi(\boldsymbol{x}) - \overline{\phi(S_{c_1})}\right]\right) + \mathbb{E}_{\boldsymbol{x}}\mathbb{E}_S\left[\phi(\boldsymbol{x}) - \overline{\phi(S_{c_1})}\right]^\top\mathbb{E}_{\boldsymbol{x}}\mathbb{E}_S[\phi(\boldsymbol{x}) - \overline{\phi(S_{c_1})}]$$

$$= \mathrm{Tr}(\Sigma_{c_1}) + \frac{1}{K}\mathrm{Tr}(\Sigma_{c_1}). \tag{27}$$

From $(i)$ and $(ii)$, and equation 19

$$\mathbb{E}_{S\sim\mathcal{D}^{\otimes 2k}}\mathbb{E}_{\boldsymbol{x}\sim\mathcal{D}_{c_1}}[\alpha] = \frac{1}{K}\left(\mathrm{Tr}\left(\Sigma_{c_2}\right) - \mathrm{Tr}\left(\Sigma_{c_1}\right)\right) + \left(\boldsymbol{\mu}_{c_1} - \boldsymbol{\mu}_{c_2}\right)^\top\left(\boldsymbol{\mu}_{c_1} - \boldsymbol{\mu}_{c_2}\right). \tag{28}$$

Since $\mathbb{E}_{c_1,c_2,\boldsymbol{x},S}[\alpha] = \mathbb{E}_{c_1,c_2\sim\tau}[\mathbb{E}_{S\sim\mathcal{D}^{\otimes 2k}}\mathbb{E}_{\boldsymbol{x}\sim\mathcal{D}_{c_1}}[\alpha]]$,

$$\mathbb{E}_{c_1,c_2,\boldsymbol{x},S}[\alpha] = \mathbb{E}_{c_1,c_2\sim\tau}\left[\frac{1}{K}(\mathrm{Tr}(\Sigma_{c_2}) - \mathrm{Tr}(\Sigma_{c_1})) + \left(\boldsymbol{\mu}_{c_1} - \boldsymbol{\mu}_{c_2}\right)^\top\left(\boldsymbol{\mu}_{c_1} - \boldsymbol{\mu}_{c_2}\right)\right]$$

$$= \mathbb{E}_{c_1,c_2\sim\tau}\left[\left(\boldsymbol{\mu}_{c_1} - \boldsymbol{\mu}_{c_2}\right)^\top\left(\boldsymbol{\mu}_{c_1} - \boldsymbol{\mu}_{c_2}\right)\right]$$

$$= \mathbb{E}_{c_1,c_2\sim\tau}\left[\boldsymbol{\mu}_{c_1}^\top\boldsymbol{\mu}_{c_1} + \boldsymbol{\mu}_{c_2}^\top\boldsymbol{\mu}_{c_2} - \boldsymbol{\mu}_{c_1}^\top\boldsymbol{\mu}_{c_2} - \boldsymbol{\mu}_{c_2}^\top\boldsymbol{\mu}_{c_1}\right]$$

$$= \mathrm{Tr}\left(\Sigma\right) + \boldsymbol{\mu}^\top\boldsymbol{\mu} + \mathrm{Tr}\left(\Sigma\right) + \boldsymbol{\mu}^\top\boldsymbol{\mu} - 2\boldsymbol{\mu}^\top\boldsymbol{\mu}$$

$$= 2\mathrm{Tr}\left(\Sigma\right). \tag{29}$$

$\square$

Next we prove the following Lemma 6 related to the conditioned variance of $\alpha$.

**Lemma 6.** *Under the same notation and assumptions as Theorem 1,*

$$\mathbb{E}_{c_1,c_2}\mathrm{Var}_{\boldsymbol{x}\sim D_{c_1},S\sim\mathcal{D}^{\otimes 2N}}[\alpha] \leq \frac{4}{K}\mathbb{E}_{c\sim\tau}\mathrm{Var}\left[\|\phi(\boldsymbol{x})\|^2\right] + \frac{4}{K}\mathrm{Var}_{c\sim\tau}\left[\mathrm{Tr}(\Sigma_c)\right] + \mathrm{V}_{\mathrm{wit}}(\Sigma_\tau, \Sigma, \boldsymbol{\mu}),$$

$$\tag{30}$$

*where*

$$\mathrm{V}_{\mathrm{wit}}\left(\Sigma_\tau, \Sigma, \boldsymbol{\mu}\right) = \frac{8}{K}\left(\mathrm{Tr}(\Sigma_\tau)\right)\left(\mathrm{Tr}(\Sigma) + \boldsymbol{\mu}^\top\boldsymbol{\mu}\right) + 4\left(\mathrm{Tr}(\Sigma) + \boldsymbol{\mu}^\top\boldsymbol{\mu}\right)^2. \tag{31}$$

*Proof.* We start with the inequality between the variance of 2 random variables. We define $\mathrm{Cov}(A, B)$ as covariance of 2 random variables $A, B$.

$$\mathrm{Var}[A + B] = \mathrm{Var}[A] + \mathrm{Var}[B] + 2\mathrm{Cov}(A, B)$$

$$\leq \mathrm{Var}[A] + \mathrm{Var}[B] + 2\sqrt{\mathrm{Var}[A]\mathrm{Var}[B]}$$

$$\leq \mathrm{Var}[A] + \mathrm{Var}[B] + 2 \cdot \frac{\mathrm{Var}[A] + \mathrm{Var}[B]}{2}$$

$$= 2\mathrm{Var}[A] + 2\mathrm{Var}[B]. \tag{32}$$

For $\mathrm{Var}_{\boldsymbol{x} \sim D_{c_1}, S}[\alpha]$,

$$
\begin{aligned}
\mathrm{Var}_{\boldsymbol{x} \sim D_{c_1}, S}[\alpha] &= \mathrm{Var}\left[\left\|\phi(\boldsymbol{x}) - \overline{\phi(S_{c_2})}\right\|^2 - \left\|\phi(\boldsymbol{x}) - \overline{\phi(S_{c_1})}\right\|^2\right] \\
&= \mathrm{Var}\left[\left\|\overline{\phi(S_{c_1})}\right\|^2 - \left\|\overline{\phi(S_{c_2})}\right\|^2 - 2\phi(\boldsymbol{x})^\top\left(\overline{\phi(S_{c_2})} - \overline{\phi(S_{c_1})}\right)\right] \\
&\leq 2\mathrm{Var}\left[\left\|\overline{\phi(S_{c_1})}\right\|^2 - \left\|\overline{\phi(S_{c_2})}\right\|^2\right] \\
&\quad + 4\mathrm{Var}\left[\phi(\boldsymbol{x})^\top\left(\overline{\phi(S_{c_2})} - \overline{\phi(S_{c_1})}\right)\right] \quad (\because equation\ 32) \\
&= 2\mathrm{Var}\left[\left\|\overline{\phi(S_{c_1})}\right\|^2\right] + 2\mathrm{Var}\left[\left\|\overline{\phi(S_{c_2})}\right\|^2\right] + 4\mathrm{Var}\left[\phi(\boldsymbol{x})^\top(\overline{\phi(S_{c_2})} - \overline{\phi(S_{c_1})})\right].
\end{aligned}
$$
$$(33)$$

From 3rd line to 4th line we deompose the variance of $\left\|\overline{\phi(S_{c_1})}\right\|^2 - \left\|\overline{\phi(S_{c_2})}\right\|^2$ use the independence of $\overline{\phi(S_{c_1})}$ and $\overline{\phi(S_{c_2})}$ with their class given. Next we focus on $\mathrm{Var}\left[\phi(\boldsymbol{x})^\top(\overline{\phi(S_{c_2})} - \overline{\phi(S_{c_1})})\right]$.

$$
\begin{aligned}
\mathrm{Var}\left[\phi(\boldsymbol{x})^\top(\overline{\phi(S_{c_2})} - \overline{\phi(S_{c_1})})\right] &= \mathbb{E}\left[\left(\phi(\boldsymbol{x})^\top\left(\overline{\phi(S_{c_2})} - \overline{\phi(S_{c_1})}\right)\right)^2\right] \\
&\quad - \left(\mathbb{E}\left[\phi(\boldsymbol{x})\right]^\top \mathbb{E}\left[\left(\overline{\phi(S_{c_2})} - \overline{\phi(S_{c_1})}\right)\right]\right)^2 \\
&\leq \mathbb{E}\left[\left(\|\phi(\boldsymbol{x})\|^2\left\|\overline{\phi(S_{c_2})} - \overline{\phi(S_{c_1})}\right\|\right)^2\right] - (\boldsymbol{\mu}_{c_1}{}^\top(\boldsymbol{\mu}_{c_2} - \boldsymbol{\mu}_{c_1}))^2 \\
&= \mathbb{E}\left[\|\phi(\boldsymbol{x})\|^2\right] \mathbb{E}\left[\left\|\overline{\phi(S_{c_2})} - \overline{\phi(S_{c_1})})\right\|^2\right] - (\boldsymbol{\mu}_{c_1}^\top(\boldsymbol{\mu}_{c_2} - \boldsymbol{\mu}_{c_1}))^2.
\end{aligned}
$$
$$(34)$$

From the 2nd line to the 3rd line we use Cauchy–Schwarz inequality. For $\mathbb{E}\left[\left\|\overline{\phi(S_{c_2})} - \overline{\phi(S_{c_1})}\right\|^2\right]$, with equation 20

$$
\mathbb{E}\left[\left\|\overline{\phi(S_{c_2})} - \overline{\phi(S_{c_1})}\right\|^2\right] = \frac{1}{K}\left(\mathrm{Tr}\left(\Sigma_{c_1}\right) + \mathrm{Tr}\left(\Sigma_{c_2}\right)\right) + \left(\boldsymbol{\mu}_{c_2} - \boldsymbol{\mu}_{c_1}\right)^\top\left(\boldsymbol{\mu}_{c_2} - \boldsymbol{\mu}_{c_1}\right). \quad (35)
$$

Thus $\mathbb{E}_{c_1,c_2\sim\tau}\left[\mathbb{E}_{\boldsymbol{x}\sim\mathcal{D}_{c_1},S}\left[\|\phi(\boldsymbol{x})\|^2\right]\mathbb{E}_{\boldsymbol{x}\sim\mathcal{D}_{c_1},S}\left[\left\|\overline{\phi(S_{c_2})}-\overline{\phi(S_{c_1})}\right\|^2\right]\right]$ is calculated as follows.

$$
\mathbb{E}_{c_1,c_2\sim\tau}\left[\mathbb{E}_{\boldsymbol{x}\sim\mathcal{D}_{c_1},S}\left[\|\phi(\boldsymbol{x})\|^2\right]\mathbb{E}_{\boldsymbol{x}\sim\mathcal{D}_{c_1},S}\left[\left\|\overline{\phi(S_{c_2})}-\overline{\phi(S_{c_1})}\right\|^2\right]\right]
$$

$$
=\mathbb{E}_{c_1,c_2\sim\tau}\left[\left(\mathrm{Tr}(\Sigma_{c_1})+\boldsymbol{\mu}_{c_1}^\top\boldsymbol{\mu}_{c_1}\right)\left(\frac{1}{K}\left(\mathrm{Tr}(\Sigma_{c_1})+\mathrm{Tr}(\Sigma_{c_2})\right)+(\boldsymbol{\mu}_{c_2}-\boldsymbol{\mu}_{c_1})^\top(\boldsymbol{\mu}_{c_2}-\boldsymbol{\mu}_{c_1})\right)\right]
$$

$$
=\mathbb{E}_{c_1,c_2\sim\tau}\left[\frac{1}{K}\left(\mathrm{Tr}\left(\Sigma_{c_1}\right)^2+\mathrm{Tr}(\Sigma_{c_1})\,\mathrm{Tr}\left(\Sigma_{c_2}\right)\right)\right]
$$

$$
+\mathbb{E}_{c_1,c_2\sim\tau}\left[\frac{2}{K}\left(\mathrm{Tr}\left(\Sigma_\tau\right)\right)\boldsymbol{\mu}_{c_1}^\top\boldsymbol{\mu}_{c_1}+\boldsymbol{\mu}_{c_1}^\top\boldsymbol{\mu}_{c_1}\left(\boldsymbol{\mu}_{c_2}-\boldsymbol{\mu}_{c_1}\right)^\top\left(\boldsymbol{\mu}_{c_2}-\boldsymbol{\mu}_{c_1}\right)\right]
$$

$$
=\frac{1}{K}\left(\mathbb{E}_{c_1,c_2\sim\tau}\left[\mathrm{Tr}(\Sigma_{c_1})^2\right]+\mathbb{E}_{c_1,c_2\sim\tau}\left[\mathrm{Tr}(\Sigma_{c_1})\right]^2\right)
$$

$$
+\frac{2}{K}\left(\mathrm{Tr}\left(\Sigma_\tau\right)\right)\left(\mathrm{Tr}\left(\Sigma\right)+\boldsymbol{\mu}^\top\boldsymbol{\mu}\right)+\mathbb{E}\left[\boldsymbol{\mu}_{c_1}^\top\boldsymbol{\mu}_{c_1}\left(\boldsymbol{\mu}_{c_2}-\boldsymbol{\mu}_{c_1}\right)^\top(\boldsymbol{\mu}_{c_2}-\boldsymbol{\mu}_{c_1})\right]
$$

$$
=\frac{1}{K}\mathrm{Var}_{c\sim\tau}\left[\mathrm{Tr}\left(\Sigma_c\right)\right]
$$

$$
+\frac{2}{K}\mathrm{Tr}\left(\Sigma_\tau\right)^2+\frac{2}{K}\left(\mathrm{Tr}\left(\Sigma_\tau\right)\right)\left(\mathrm{Tr}\left(\Sigma\right)+\boldsymbol{\mu}^\top\boldsymbol{\mu}\right)+\mathbb{E}_{c_1,c_2}\left[\boldsymbol{\mu}_{c_1}^\top\boldsymbol{\mu}_{c_1}\left(\boldsymbol{\mu}_{c_2}-\boldsymbol{\mu}_{c_1}\right)^\top\left(\boldsymbol{\mu}_{c_2}-\boldsymbol{\mu}_{c_1}\right)\right]
$$

$$
=\frac{1}{K}\mathrm{Var}_{c\sim\tau}\left[\mathrm{Tr}(\Sigma_c)\right]
$$

$$
+\frac{2}{K}\mathrm{Tr}(\Sigma_\tau)\left(\mathrm{Tr}(\Sigma_\tau)+\mathrm{Tr}(\Sigma)+\boldsymbol{\mu}^\top\boldsymbol{\mu}\right)+\mathbb{E}_{c_1,c_2}\left[\boldsymbol{\mu}_{c_1}^\top\boldsymbol{\mu}_{c_1}\left(\boldsymbol{\mu}_{c_2}-\boldsymbol{\mu}_{c_1}\right)^\top\left(\boldsymbol{\mu}_{c_2}-\boldsymbol{\mu}_{c_1}\right)\right].
$$
(36)

Now we take into account the term $-(\boldsymbol{\mu}_{c_1}\top(\boldsymbol{\mu}_{c_2}-\boldsymbol{\mu}_{c_1}))^2$ in equation 34,

$$
\mathbb{E}_{c_1,c_2}\left[\boldsymbol{\mu}_{c_1}^\top\boldsymbol{\mu}_{c_1}(\boldsymbol{\mu}_{c_2}-\boldsymbol{\mu}_{c_1})^\top(\boldsymbol{\mu}_{c_2}-\boldsymbol{\mu}_{c_1})-(\boldsymbol{\mu}_{c_1}^\top(\boldsymbol{\mu}_{c_2}-\boldsymbol{\mu}_{c_1}))^2\right]
$$

$$
=\mathbb{E}_{c_1,c_2}\left[\boldsymbol{\mu}_{c_1}^\top\boldsymbol{\mu}_{c_1}\boldsymbol{\mu}_{c_2}^\top\boldsymbol{\mu}_{c_2}-(\boldsymbol{\mu}_{c_1}^\top\boldsymbol{\mu}_{c_2})^2\right]
$$

$$
\le\mathbb{E}_{c_1,c_2}\left[\boldsymbol{\mu}_{c_1}^\top\boldsymbol{\mu}_{c_1}\boldsymbol{\mu}_{c_2}^\top\boldsymbol{\mu}_{c_2}\right]
$$

$$
=\left(\mathrm{Tr}(\Sigma)+\boldsymbol{\mu}^\top\boldsymbol{\mu}\right)^2.
$$
(37)

Thus $\mathbb{E}_{c_1,c_2}\left[\mathrm{Var}[\phi(\boldsymbol{x})^\top(\overline{\phi(S_{c_2})}-\overline{\phi(S_{c_1})})]\right]$ is calculated as follows.

$$
\mathbb{E}_{c_1,c_2}\left[\mathrm{Var}[\phi(\boldsymbol{x})^\top(\overline{\phi(S_{c_2})}-\overline{\phi(S_{c_1})})]\right]
$$
(38)

$$
=\frac{1}{K}\mathrm{Var}_{c\sim\tau}\left[\mathrm{Tr}(\Sigma_c)\right]+\frac{2}{K}\mathrm{Tr}(\Sigma_\tau)\left(\mathrm{Tr}(\Sigma_\tau)+\mathrm{Tr}(\Sigma)+\boldsymbol{\mu}^\top\boldsymbol{\mu}\right)+(\mathrm{Tr}(\Sigma)+\boldsymbol{\mu}^\top\boldsymbol{\mu})^2.
$$
(39)

Regarding $\left\|\overline{\phi(S_c)}\right\|^2$, since the function computing square norm is convex, next equation holds with $D$-dimensional Jensen's inequality (Perlman, 1974):

$$
\left\|\overline{\phi(S_c)}\right\|^2=\left\|\frac{1}{K}\sum_{\substack{i=0\\x\in S_c}}\phi(\boldsymbol{x})\right\|^2
$$

$$
\le\frac{1}{K}\left\|\sum_{\substack{i=0\\x\in S_c}}\phi(\boldsymbol{x})\right\|.
$$
(40)

Combining equation 33, equation 38, and equation 40 we obtain

$$
\begin{aligned}
&\mathbb{E}_{c_1,c_2}\mathrm{Var}_{\boldsymbol{x}\sim D_{c_1},S\sim\mathcal{D}^{\otimes 2N}}[\alpha] \\
&= \frac{4}{K}\mathbb{E}_{c\sim\tau}\mathrm{Var}_{\boldsymbol{x}\sim\mathcal{D}_c}\left[\|\phi(\boldsymbol{x})\|^2\right] + \frac{4}{K}\mathrm{Var}_{c\sim\tau}\left[\mathrm{Tr}(\Sigma_c)\right] \\
&\quad + \frac{8}{K}\mathrm{Tr}(\Sigma_\tau)\left(\mathrm{Tr}(\Sigma_\tau)+\mathrm{Tr}(\Sigma)+\boldsymbol{\mu}^\top\boldsymbol{\mu}\right) + 4\left(\mathrm{Tr}(\Sigma)+\boldsymbol{\mu}^\top\boldsymbol{\mu}\right)^2.
\end{aligned}
\tag{41}
$$

$\square$

To complete the proof of Theorem 1, we calculate $\mathbb{E}_{c_1,c_2}\mathbb{E}_{\boldsymbol{x}\sim\mathcal{D}_{c_1},S\sim\mathcal{D}^{\otimes 2K}}[\alpha]^2$.

$$
\mathbb{E}_{c_1,c_2}\mathbb{E}_{\boldsymbol{x}\sim\mathcal{D}_{c_1},S\sim\mathcal{D}^{\otimes 2K}}[\alpha]^2 \tag{42}
$$
$$
= E_{c_1,c_2}\left[\left(\frac{1}{K}(\mathrm{Tr}(\Sigma_{c_1})-\mathrm{Tr}(\Sigma_{c_2})) + (\boldsymbol{\mu}_{c_1}-\boldsymbol{\mu}_{c_2})^\top(\boldsymbol{\mu}_{c_1}-\boldsymbol{\mu}_{c_2})\right)^2\right]
$$
$$
= \mathbb{E}_{c_1,c_2}\left[\left(\frac{1}{K}(\mathrm{Tr}(\Sigma_{c_1})-\mathrm{Tr}(\Sigma_{c_2}))\right)^2\right] + \mathbb{E}_{c_1,c_2}\left[\left((\boldsymbol{\mu}_{c_1}-\boldsymbol{\mu}_{c_2})^\top(\boldsymbol{\mu}_{c_1}-\boldsymbol{\mu}_{c_2})\right)^2\right]
$$
$$
\quad + 2\mathbb{E}_{c_1,c_2}\left[\left(\frac{1}{K}(\mathrm{Tr}(\Sigma_{c_1})-\mathrm{Tr}(\Sigma_{c_2}))\right)\left((\boldsymbol{\mu}_{c_1}-\boldsymbol{\mu}_{c_2})^\top(\boldsymbol{\mu}_{c_1}-\boldsymbol{\mu}_{c_2})\right)\right]
$$
$$
= \mathrm{Var}_{c_1,c_2\sim\tau}\left[\frac{1}{K}(\mathrm{Tr}(\Sigma_{c_1})-\mathrm{Tr}(\Sigma_{c_2}))\right] + \left(\mathbb{E}_{c_1,c_2\sim\tau}\left[\frac{1}{K}(\mathrm{Tr}(\Sigma_{c_1})-\mathrm{Tr}(\Sigma_{c_2}))\right]\right)^2
$$
$$
\quad + \mathbb{E}_{c_1,c_2}\left[\left((\boldsymbol{\mu}_{c_1}-\boldsymbol{\mu}_{c_2})^\top(\boldsymbol{\mu}_{c_1}-\boldsymbol{\mu}_{c_2})\right)^2\right]
$$
$$
= \frac{2}{K^2}\mathrm{Var}_{c\sim\tau}\left[\mathrm{Tr}(\Sigma_c)\right] + \mathbb{E}_{c_1,c_2}\left[\left((\boldsymbol{\mu}_{c_1}-\boldsymbol{\mu}_{c_2})^\top(\boldsymbol{\mu}_{c_1}-\boldsymbol{\mu}_{c_2})\right)^2\right]. \tag{43}
$$

From 2nd line to 3rd line, we use the symmetry of the last term with respect to $c_1$ and $c_2$ and erase the term. Combining equation 13, Lemma 5, Lemma 6, and equation 42, we obtain the bound.

## A.5 Theorem 1 with $N$-way Classification

The upper bound on the risk of $N$-way prototype classifier is as follows.

**Theorem 7.** *Let operation of binary class prototype classifier $\mathcal{M}$ as defined in equation 1. Then for $\overline{\phi(S_c)} = \frac{1}{K}\Sigma_{\boldsymbol{x}\in S_c}\phi(\boldsymbol{x})$, $\mu_c = \mathbb{E}_{\boldsymbol{x}\sim\mathcal{D}_c}[\phi(\boldsymbol{x})]$, $\Sigma_c = \mathbb{E}_{\boldsymbol{x}\sim\mathcal{D}_c}[(\phi(\boldsymbol{x})-\mu_c)(\phi(\boldsymbol{x})-\mu_c)^\top]$, $\mu = \mathbb{E}_{c\sim\tau}[\mu_c]$, $\Sigma = \mathbb{E}_{c\sim\tau}[(\mu_c-\mu)(\mu_c-\mu)^\top]$, $\mathbb{E}_{c\sim\tau}[\Sigma_c] = \Sigma_\tau$, if $\phi(\boldsymbol{x})$ has its fourth moment, miss classification risk of binary class prototype classifier $\mathrm{R}_\mathcal{M}$ satisfy*

$$
\begin{aligned}
&\mathrm{R}(\mathcal{M}(\phi,\boldsymbol{x},\{S_i\}_{i=1}^N),y) \\
&\leq N-1-\sum_{\substack{c=1\\c\neq y}}^N \frac{4(\mathrm{Tr}(\Sigma))^2}{\mathbb{E}\mathrm{V}[h_{L2}(\phi(\boldsymbol{x}))]+\mathrm{V}_{\mathrm{Tr}}(\Sigma_y)+\mathrm{V}_{\mathrm{wit}}(\Sigma_\tau,\Sigma,\boldsymbol{\mu})+\mathbb{E}\,\mathrm{dist}_{\mathrm{L2}}^2(\boldsymbol{\mu}_y,\boldsymbol{\mu}_c)},
\end{aligned}
\tag{44}
$$

*where*

$$
\mathbb{E}\mathrm{V}[h_{L2}(\phi(\boldsymbol{x}))] = \frac{4}{K}\mathbb{E}_{y\sim\tau}\left[\mathrm{Var}_{\boldsymbol{x}_c\sim\mathcal{D}_c}\left[\|\phi(\boldsymbol{x})\|^2\right]\right],
$$
$$
\mathrm{V}_{\mathrm{Tr}}(\Sigma_y) = \left(\frac{4}{K}+\frac{2}{K^2}\right)\mathrm{Var}_{c\sim\tau}\left[\mathrm{Tr}(\Sigma_c)\right],
$$
$$
\mathrm{V}_{\mathrm{wit}}(\Sigma_\tau,\Sigma,\boldsymbol{\mu}) = \frac{8}{K}\mathrm{Tr}(\Sigma_\tau)\left(\mathrm{Tr}(\Sigma_\tau)+\mathrm{Tr}(\Sigma)+\boldsymbol{\mu}^\top\boldsymbol{\mu}\right)+4(\mathrm{Tr}(\Sigma)+\boldsymbol{\mu}^\top\boldsymbol{\mu})^2,
$$
$$
\mathbb{E}\,\mathrm{dist}_{\mathrm{L2}}^2(\boldsymbol{\mu}_y,\boldsymbol{\mu}_c) = \mathbb{E}_{y,c}\left[\left((\boldsymbol{\mu}_{c_y}-\boldsymbol{\mu}_c)^\top(\boldsymbol{\mu}_y-\boldsymbol{\mu}_c)\right)^2\right].
$$

Table 3: Classification accuracies on *mini*ImageNet of ProtoNet , linear evaluation methods (Chen et al., 2019), and ours. The best performing methods and any other runs within 95% confidence margin are in bold.

| Method | ResNet12 | | ResNet18 | |
|---|---|---|---|---|
| | 1-shot | 5-shot | 1-shot | 5-shot |
| ProtoNet | $53.48 \pm 0.89\%$ | $73.56 \pm 0.65\%$ | $56.26 \pm 0.85\%$ | $74.02 \pm 0.65\%$ |
| B@FT | $54.54 \pm 0.80\%$ | $76.50 \pm 0.62\%$ | $55.41 \pm 0.82\%$ | $\mathbf{76.95 \pm 0.61}\%$ |
| B+@FT | $56.33 \pm 0.81\%$ | $74.62 \pm 0.65\%$ | $55.07 \pm 0.81\%$ | $74.15 \pm 0.66\%$ |
| B@CL2 | $\mathbf{58.66 \pm 0.83\%}$ | $75.98 \pm 0.62\%$ | $\mathbf{57.67 \pm 0.83}\%$ | $70.78 \pm 0.67\%$ |
| B+@CL2 | $57.50 \pm 0.81\%$ | $74.00 \pm 0.61\%$ | $\mathbf{57.00 \pm 0.64}\%$ | $74.06 \pm 0.61\%$ |
| B | $46.36 \pm 0.58\%$ | $73.97 \pm 0.62\%$ | $43.86 \pm 0.80\%$ | $72.36 \pm 0.92\%$ |
| B@L2-N | $57.18 \pm 0.80\%$ | $\mathbf{77.12 \pm 0.44\%}$ | $56.57 \pm 0.80\%$ | $\mathbf{76.44 \pm 0.61}\%$ |
| B@V-N | – | $63.55 \pm 0.49\%$ | – | $62.78 \pm 1.02\%$ |
| B@V-N+L2-N | – | $65.42 \pm 0.64\%$ | – | $64.09 \pm 0.67\%$ |
| B@LDA | – | $73.75 \pm 0.64\%$ | – | $73.54 \pm 0.85\%$ |
| B@LDA+L2-N | – | $74.80 \pm 0.64\%$ | – | $73.70 \pm 0.62\%$ |
| B@EST | $51.28 \pm 0.88\%$ | $72.80 \pm 0.64\%$ | $44.19 \pm 0.82\%$ | $72.99 \pm 0.63\%$ |
| B@EST+L2-N | $\mathbf{58.00 \pm 0.86\%}$ | $\mathbf{76.90 \pm 0.62\%}$ | $56.39 \pm 0.79\%$ | $\mathbf{76.24 \pm 0.64}\%$ |
| B+ | $41.18 \pm 0.76\%$ | $73.97 \pm 0.66\%$ | $36.80 \pm 0.76\%$ | $63.76 \pm 0.71\%$ |
| B+@L2-N | $57.96 \pm 0.83\%$ | $75.38 \pm 0.61\%$ | $\mathbf{57.21 \pm 0.83}\%$ | $74.89 \pm 0.65\%$ |
| B+@V-N | – | $55.96 \pm 0.63\%$ | – | $54.31 \pm 0.64\%$ |
| B+@V-N+L2-N | – | $56.20 \pm 0.64\%$ | – | $55.42 \pm 0.68\%$ |
| B+@LDA | – | $69.60 \pm 0.70\%$ | – | $74.38 \pm 0.65\%$ |
| B+@LDA+L2-N | – | $74.34 \pm 0.63\%$ | – | $74.38 \pm 0.63\%$ |
| B+@EST | $47.11 \pm 0.81\%$ | $69.01 \pm 0.63\%$ | $47.21 \pm 0.77\%$ | $69.11 \pm 0.64\%$ |
| B+@EST+L2-N | $\mathbf{58.32 \pm 0.81\%}$ | $\mathbf{77.49 \pm 0.63\%}$ | $\mathbf{57.00 \pm 0.75}\%$ | $\mathbf{77.13 \pm 0.64}\%$ |

*Proof.* Let $x, y$ be the input and its class of a query data. We define $\alpha_c$ by $\alpha_c = \left\| \phi(\boldsymbol{x}) - \overline{\phi(S_c)} \right\|^2 - \left\| \phi(\boldsymbol{x}) - \overline{\phi(S_y)} \right\|^2$. Then a prototype classifier miss-classify a class of input $x$, $\hat{y}$, if $\exists c \in [1, N], c \neq y, \alpha_c < 0$. Hence: $\mathrm{R}_{\mathcal{M}}(\phi) = \mathrm{Pr}(\bigcup_{\substack{c=1 \\ c \neq y}}^{N} \alpha_c < 0)$

By Frechet's inequality, next equation holds:

$$\mathrm{R}_{\mathcal{M}}(\phi) \leq \sum_{\substack{c=1 \\ c \neq y}}^{N} \mathrm{Pr}(\alpha_i < 0).$$

Noting that Theorem 1 can be applied to each term in the summation and then we obtain Theorem 7 □

## A.6 DETAILED PERFORMANCE RESULTS

We show the detailed performance results in this section with 95% confidence margin. We add the two future normalization methods, variance normalization before $L_2$-normalization (V-N+L2-N) and LDA before $L_2$-normalization (LDA+L2-N) to the tables. Table 3 and Table 4 show the performance on *mini*Imagenet and *tiered*Imagenet; table 5 and table 6 show the performance on CIFARFS and FC100.

## A.7 THE COMPARISON OF THE RATIO OF THE BETWEEN-CLASS VARIANCE TO THE WITHIN-CLASS VARIANCE BEFORE AND AFTER APPLYING $L_2$-NORM

We show the ratio of the between-class variance to the within-class variance before and after applying $L_2$-norm in Figure 4. We calculated the ratio of the between-class variance to each class's variance and averaged over the test classes of each dataset. Although $L_2$-normalization is expected to reduce $\mathrm{Tr}(\Sigma)$ rather than $\Sigma_c$ according to 4.4, the figure shows $L_2$-norm marginaly reduces the ratio in CIFARFS dataset and the ratio does not changed so much in the other dataset.

Table 4: Classification accuracies on *tiered*ImageNet of ProtoNet , linear evaluation methods (Chen et al., 2019), and ours. The best performing methods and any other runs within 95% confidence margin are in bold.

| Method | ResNet12 | | ResNet18 | |
|---|---|---|---|---|
| | 1-shot | 5-shot | 1-shot | 5-shot |
| ProtoNet | $55.40 \pm 0.98\%$ | $77.67 \pm 0.70\%$ | $60.50 \pm 1.01\%$ | $81.40 \pm 0.68\%$ |
| B@FT | $61.67 \pm 0.92\%$ | $\mathbf{81.62 \pm 0.64\%}$ | $63.38 \pm 0.91\%$ | $\mathbf{83.18 \pm 0.64\%}$ |
| B+@FT | $63.02 \pm 0.91\%$ | $\mathbf{81.07 \pm 0.69\%}$ | $64.20 \pm 0.92\%$ | $81.62 \pm 0.69\%$ |
| B@CL2 | $\mathbf{64.88 \pm 0.86\%}$ | $80.42 \pm 0.64\%$ | $\mathbf{65.26 \pm 0.88\%}$ | $81.63 \pm 0.64\%$ |
| B+@CL2 | $63.31 \pm 0.91\%$ | $79.19 \pm 0.68\%$ | $65.67 \pm 0.91\%$ | $81.41 \pm 0.68\%$ |
| B | $50.60 \pm 0.87\%$ | $78.10 \pm 0.67\%$ | $56.16 \pm 0.89\%$ | $80.33 \pm 0.66\%$ |
| B@L2-N | $\mathbf{64.04 \pm 0.89\ \%}$ | $\mathbf{81.73 \pm 0.66\%}$ | $\mathbf{65.19 \pm 0.87\%}$ | $82.93 \pm 0.66\%$ |
| B@V-N | – | $66.08 \pm 0.69\%$ | – | $75.56 \pm 0.75\%$ |
| B@V-N+L2-N | – | $66.85 \pm 0.70\%$ | – | $76.66 \pm 0.67\%$ |
| B@LDA | – | $77.44 \pm 0.74\%$ | – | $80.85 \pm 0.67\%$ |
| B@LDA+L2-N | – | $81.01 \pm 0.69\%$ | – | $81.48 \pm 0.68\%$ |
| B@EST | $53.90 \pm 0.94\%$ | $78.09 \pm 0.67\%$ | $57.12 \pm 0.93\%$ | $80.59 \pm 0.64\%$ |
| B@EST+L2-N | $\mathbf{64.54 \pm 0.91\%}$ | $\mathbf{81.40 \pm 0.64\%}$ | $\mathbf{64.71 \pm 0.91\%}$ | $\mathbf{83.24 \pm 0.67\%}$ |
| B+ | $46.52 \pm 0.87\%$ | $73.85 \pm 0.70\%$ | $48.27 \pm 0.91\%$ | $75.87 \pm 0.71\%$ |
| B+@L2-N | $\mathbf{64.96 \pm 0.90\%}$ | $81.08 \pm 0.69\%$ | $\mathbf{65.40 \pm 0.94\%}$ | $82.49 \pm 0.68\%$ |
| B+@V-N | – | $65.19 \pm 0.79\%$ | – | $64.33 \pm 0.77\%$ |
| B+@V-N+L2-N | – | $66.57 \pm 0.74\%$ | – | $67.15 \pm 0.74\%$ |
| B+@LDA | – | $72.88 \pm 0.74\%$ | – | $75.55 \pm 0.77\%$ |
| B+@LDA+L2-N | – | $79.57 \pm 0.72\%$ | – | $81.98 \pm 0.67\%$ |
| B+@EST | $52.01 \pm 0.90\%$ | $74.26 \pm 0.70\%$ | $53.49 \pm 0.90\%$ | $75.38 \pm 0.71\%$ |
| B+@EST+L2-N | $57.63 \pm 0.88\%$ | $\mathbf{80.99 \pm 0.72\%}$ | $60.87 \pm 1.01\%$ | $81.80 \pm 0.67\%$ |

Table 5: Classification accuracies on CIFARFS of ProtoNet, linear evaluation methods (Chen et al., 2019), and ours. The best performing methods and any other runs within 95% confidence margin are in bold.

| Method | ResNet12 | | ResNet18 | |
|---|---|---|---|---|
| | 1-shot | 5-shot | 1-shot | 5-shot |
| ProtoNet | $58.65 \pm 1.07\%$ | $75.33 \pm 0.73\%$ | $62.05 \pm 0.97\%$ | $\mathbf{78.25 \pm 0.71\ \%}$ |
| B@FT | $55.97 \pm 0.86\%$ | $\mathbf{76.50 \pm 0.74\%}$ | $56.46 \pm 0.84\%$ | $77.43 \pm 0.66\%$ |
| B+@FT | $\mathbf{61.08 \pm 0.95\%}$ | $76.15 \pm 0.75\%$ | $61.34 \pm 0.93\%$ | $\mathbf{77.86 \pm 0.71\%}$ |
| B@CL2 | $58.61 \pm 0.86\%$ | $74.73 \pm 0.74\%$ | $58.09 \pm 0.84\%$ | $76.52 \pm 0.71\%$ |
| B+@CL2 | $\mathbf{61.59 \pm 0.91\%}$ | $76.10 \pm 0.73\%$ | $\mathbf{63.17 \pm 0.94\%}$ | $\mathbf{77.49 \pm 0.72\%}$ |
| B | $44.12 \pm 0.89\%$ | $72.47 \pm 0.73\%$ | $47.62 \pm 0.84\%$ | $75.66 \pm 0.71\%$ |
| B@L2-N | $59.43 \pm 0.90\%$ | $\mathbf{77.45 \pm 0.70\ \%}$ | $58.51 \pm 0.86\%$ | $77.43 \pm 0.69\%$ |
| B@V-N | – | $44.66 \pm 1.44\%$ | – | $67.72 \pm 0.72\%$ |
| B@V-N+L2-N | – | $55.91 \pm 0.70\%$ | – | $67.70 \pm 0.69\%$ |
| B@LDA | – | $72.61 \pm 0.70\%$ | – | $75.25 \pm 0.67\%$ |
| B@LDA+ L2-N | – | $75.73 \pm 0.71\%$ | – | $77.12 \pm 0.71\%$ |
| B@EST | $52.62 \pm 0.91\%$ | $72.80 \pm 0.71\%$ | $55.57 \pm 0.90\%$ | $75.10 \pm 0.71\%$ |
| B@EST+L2-N | $\mathbf{60.70 \pm 0.95\ \%}$ | $\mathbf{77.04 \pm 0.70\ \%}$ | $61.14 \pm 0.96\%$ | $\mathbf{77.48 \pm 0.68\%}$ |
| B+ | $45.73 \pm 0.88\%$ | $73.97 \pm 0.76\%$ | $44.42 \pm 0.89\%$ | $70.85 \pm 0.76\%$ |
| B+@L2-N | $\mathbf{61.07 \pm 0.93\ \%}$ | $76.71 \pm 0.74\%$ | $\mathbf{63.48 \pm 0.94\%}$ | $\mathbf{77.86 \pm 0.71\%}$ |
| B+@V-N | – | $57.20 \pm 0.70\%$ | – | $57.16 \pm 0.72\%$ |
| B+@V-N+L2-N | – | $57.59 \pm 0.69\%$ | – | $57.24 \pm 0.73\%$ |
| B+@LDA | – | $68.58 \pm 0.74\%$ | – | $69.86 \pm 0.74\%$ |
| B+@LDA+L2-N | – | $76.09 \pm 0.75\%$ | – | $69.86 \pm 0.74\%$ |
| B+@EST | $50.18 \pm 0.93\%$ | $70.32 \pm 0.79\%$ | $49.82 \pm 0.94\%$ | $71.07 \pm 0.70\%$ |
| B+@EST+L2-N | $59.83 \pm 0.98\%$ | $76.32 \pm 0.72\%$ | $\mathbf{63.00 \pm 0.94\%}$ | $\mathbf{77.99 \pm 0.71\%}$ |

Table 6: Classification accuracies on FC100 of ProtoNet, linear evaluation methods (Chen et al., 2019), and ours. The best performing methods and any other runs within 95% confidence margin are in bold.

| Method | ResNet12 | | ResNet18 | |
|---|---|---|---|---|
| | 1-shot | 5-shot | 1-shot | 5-shot |
| ProtoNet | $35.56 \pm 0.77\%$ | $51.12 \pm 0.71\%$ | $36.02 \pm 0.70\%$ | $51.02 \pm 0.72\%$ |
| B@FT | $39.72 \pm 0.68\%$ | $\mathbf{56.04 \pm 0.76\%}$ | $40.06 \pm 0.68\%$ | $57.04 \pm 0.71\%$ |
| B+@FT | $36.01 \pm 0.64\%$ | $50.73 \pm 0.72\%$ | $36.93 \pm 0.70\%$ | $50.41 \pm 0.73\%$ |
| B@CL2 | $41.58 \pm 0.74\%$ | $55.82 \pm 0.76\%$ | $41.51 \pm 0.72\%$ | $56.44 \pm 0.74\%$ |
| B+@CL2 | $37.51 \pm 0.71\%$ | $50.38 \pm 0.72\%$ | $38.30 \pm 0.74\%$ | $51.06 \pm 0.70\%$ |
| B | $31.60 \pm 0.61\%$ | $52.50 \pm 0.74\%$ | $35.90 \pm 0.61\%$ | $55.20 \pm 0.77\%$ |
| B@L2-N | $40.34 \pm 0.71\%$ | $\mathbf{56.61 \pm 0.71\%}$ | $40.49 \pm 0.71\%$ | $57.71 \pm 0.76\%$ |
| B@V-N | $-$ | $41.99 \pm 0.63\%$ | $-$ | $49.69 \pm 0.67\%$ |
| B@V-N+L2-N | $-$ | $42.03 \pm 0.65\%$ | $-$ | $49.99 \pm 0.70\%$ |
| B@LDA | $-$ | $52.90 \pm 0.72\%$ | $-$ | $55.44 \pm 0.72\%$ |
| B@LDA+L2-N | $-$ | $55.12 \pm 0.70\%$ | $-$ | $56.32 \pm 0.75\%$ |
| B@EST | $44.45 \pm 0.88\%$ | $52.97 \pm 0.81\%$ | $47.49 \pm 0.91\%$ | $55.68 \pm 0.74\%$ |
| B@EST+L2-N | $\mathbf{47.57 \pm 0.83\%}$ | $\mathbf{56.96 \pm 0.77\%}$ | $\mathbf{50.13 \pm 0.91\%}$ | $\mathbf{59.94 \pm 0.74\%}$ |
| B+ | $29.72 \pm 0.57\%$ | $46.85 \pm 0.67\%$ | $30.76 \pm 0.62\%$ | $47.62 \pm 0.69\%$ |
| B+@L2-N | $37.16 \pm 0.67\%$ | $51.10 \pm 0.71\%$ | $38.55 \pm 0.72\%$ | $50.15 \pm 0.71\%$ |
| B+@V-N | $-$ | $36.76 \pm 0.59\%$ | $-$ | $40.66 \pm 0.64\%$ |
| B+@V-N+L2-N | $-$ | $36.55 \pm 0.55\%$ | $-$ | $40.16 \pm 0.61\%$ |
| B+@LDA | $-$ | $47.10 \pm 0.70\%$ | $-$ | $47.63 \pm 0.69\%$ |
| B+@LDA+L2-N | $-$ | $49.03 \pm 0.70\%$ | $-$ | $50.60 \pm 0.72\%$ |
| B+@EST | $37.60 \pm 0.78\%$ | $46.81 \pm 0.67\%$ | $39.92 \pm 0.88\%$ | $47.65 \pm 0.69\%$ |
| B+@EST+L2-N | $40.88 \pm 0.83\%$ | $50.45 \pm 0.72\%$ | $41.65 \pm 0.85\%$ | $50.74 \pm 0.70\%$ |

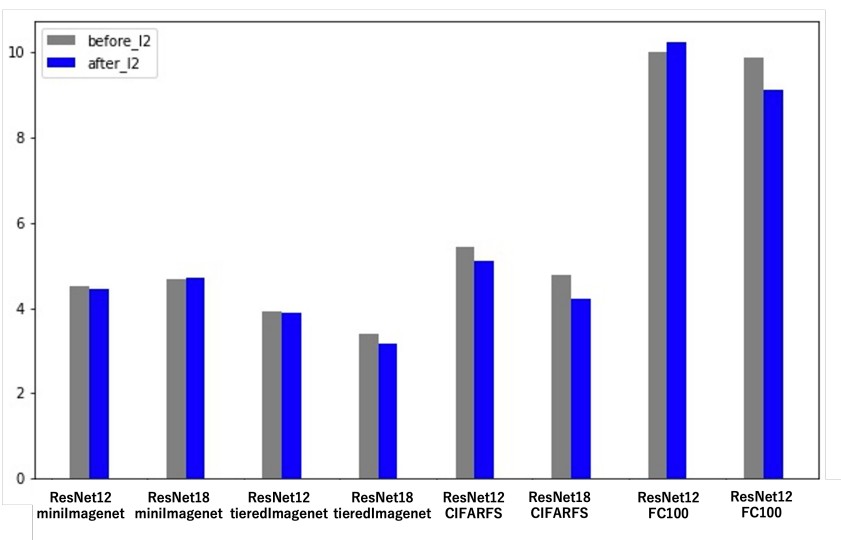

Figure 4: We plotted the ratio of the between-class variance to the within-class variance ($\frac{\mathrm{Tr}(\Sigma_c)}{\mathrm{Tr}(\Sigma)}$) before and after applying $L_2$-norm averaged over the test classes of each dataset.

