# OpenReview forum: "A Closer Look at Prototype Classifier for Few-shot Image Classification"
_ICLR.cc/2022/Conference — ICLR 2022 Submitted_

### Official Review · Reviewer_8nVE · 2021-11-02

**Correctness:** 3
**Technical Novelty And Significance:** 2
**Empirical Novelty And Significance:** 2
**Recommendation:** 6
**Confidence:** 4

**Main Review:**

The interesting aspects of the paper are as follows:
1. Theoretical analysis of protonets with relaxed constraints is interesting.
2. Evaluation on benchmarks and experimental results seems to be consistent.

However, some concerns are as follows:
1. The theoretical analysis just relax the assumption of having particular class distribution (in both class distribution and class covariance matrix), but the overall approach is marginally novel compared to Cao et al.
2. The theoretical results derived in this paper are quite intuitive though, e.g., L2 normalization of features is effective. These are quite well-known tricks. I was looking for more interesting insights which are non-trivial.
3.  Variance normalization performs worse - can you connect this with the theoretical analysis performed here?
4. How this paper addressed the second drawback, i.e., what is the optimal feature transformation, that is not clear to me. Several feature transformation methods are evaluated, however, there is no guarantee that the chosen feature transformation method is optimal.
5. The flow of the paper might be improved, specially in the experimental results section.

**Summary Of The Paper:**

In this paper, the authors have derived a generalization bound for the prototypical networks, which provides some insights towards improving protonets without finetuning using simply normalizing the feature vectors and reducing the variance of the norm. The derivation has relaxed the previous assumption of considering feature distribution and class covariance matrix to be a particular form. To justify the claim, the authors experimented on several feature transformation methods on standard few-shot benchmarks.



**Summary Of The Review:**

Overall the theoretical analysis is interesting, however the novelty seems to be marginal. The insights provided in the paper are quite intuitive  and proper connection with theory and experiments need to be explained better. I would like to see how the authors response the raised issues.

---

> ### Author Response · Authors · 2021-11-17
> **Response to Reviewer 8nVE**
>
> We would like to thank reviewer 8nVE for providing valuable feedback and raising interesting questions which we answer below.
>
> **Q1.The theoretical analysis just relax the assumption of having particular class distribution (in both class distribution and class covariance matrix), but the overall approach is marginally novel compared to Cao et al.**
>
> **A1** Our theoretical analysis has remarkable progress beyond relaxing the assumption compared to Cao et al. as follows. There are two problems in Cao’s bound when we consider a model pre-trained on cross-entropy loss. One is that the distributions of features are not guaranteed to be Gaussian and the covariance of each class conditional distribution is not the same among each other. We can confirm this from Figure 1-(b). Second is that Figure 1 suggests that reducing the variance of the norm is expected to improve the performance and we show L2-normalization indeed improves the performance in the experimental part, yet Cao’s bound cannot explain this phenomenon. Our bound overcomes these drawbacks. Further, our goal is to seek how to achieve comparable performance with fine-tuning or meta-learning when we do not want to do them. Our bound suggests the effective transformation methods which Cao’s bound cannot suggest and we claim it is important.
>
> **Q2. Are there more interesting insights which are non-trivial?**
>
> **A2** Our analysis is superior and has remarkable progress in the following two points.
> - l2-norm is indeed a well-known method, but there are no examples that theoretically show its contribution to accuracy improvement, but our generalization bound can show it.
> - When a model is pre-trained with cross-entropy loss, the features of the model have a large variance on its norm and a large generalization bound. Our generalization bound suggests that L2-normalization improves the performance of the models by reducing the variance of the norm.
>
> **Q3 Variance normalization performs worse - can you connect this with the theoretical analysis performed here?**
>
> **A3** There are two reasons that variance normalization performs worse. One is that as described in Section 4.3 and Figure 2 Left, the value of the term related to the difference of variance is relatively small compared to the variance of the norm or the term related to the ratio of the between-class variance to the within-class variance and reducing the term has little effect on reducing the bound. Second is that as described in Section 4.3, p.8, fourth paragraph, estimating the variance with few examples is unstable and inaccurate. For these two reasons, the variance normalization performance is worse.
>
> **Q4. How this paper addressed the second drawback, i.e., what is the optimal feature transformation, that is not clear to me**
>
> **A4** Since we can't define the whole space of feature transformation methods, we can't theoretically show the optimality of the selection method. However, we can provide some guidelines to decide what feature transformation methods to use when we want to improve the performance of a prototype classifier.
> We can consider from the bound that reducing the following three terms can improve performance:
>
> (a): The variance of the norm (equation 6),
>
> (b): the difference of the variance among the classes (equation 7),
>
> (c): the ratio of the between-class variance to the within-class variance (equation 8).
>
> Therefore we list the transformation methods expected to reduce (a), (b), (c) and conduct experiments. From the experimental results, the feature transformation methods listed there indeed improve the performance of the model and achieve comparable performance with fine-tuning-based methods and meta-learning-based methods.
>
> **Q5. The flow of the paper might be improved, especially in the experimental results section.**
>
> **A5** Thank you for pointing that out. We have modified Section 4.3. so that makes the flow of the experimental part clearer.

---

> > ### Comment · Reviewer_8nVE · 2021-11-22
> > **Post rebuttal comment**
> >
> > Although, the authors addressed my doubts, I still think the novelty is marginal compared to the previous work and the findings are quite intuitive. Also, reviewer 2 pointed out some errors in the theoretical analysis, which the authors addressed in the rebuttal. The authors provide detailed experiments and analysis, which is impressive, therefore, I am keeping my rating unchanged to weak accept.

---

### Official Review · Reviewer_krD7 · 2021-11-04

**Correctness:** 4
**Technical Novelty And Significance:** 3
**Empirical Novelty And Significance:** 2
**Recommendation:** 6
**Confidence:** 4

**Main Review:**

[Strengths]
- It is interesting to focus on the different loss functions in feature training (softmax loss) and testing (Prototype classifier).

- The method which works well without meta-learning and fine-tuning is practical.

- L2 normalization is well known to improve performance. The paper explains how the variance of the norm is related to the bound of a prototype classifier.

[Weakness]

The comparison of normalization is incomplete.
- The L2-N is applied only to EST. Why L2-N is not applied to V-N and LDA.
- It is not clear if B@CL2 and B+@CL2 use a prototypical classifier, or fine-tuning is conducted.

The discussion of Sec.4.4 is not completely validated. It claims that  the L2-norm normalization less influences ${\rm Tr}(\Sigma)$ than ${\rm Tr}(\Sigma_c)$. These values before and after the normalization also should be compared.

There are several unclear or incorrect descriptions.

- It is claimed that the proposed bound does not assume any specific distribution and the same covariance matrix among classes. However, it seems that the variance and mean of Gaussian distribution are used when expanding the results of one-sided Chebyshev's inequality, eg., Eq.(22). Also, p.15 Eq.(29), the first line to the second line, it seems to be assuming ${\rm Tr}(\Sigma_{c2}) = {\rm Tr}(\Sigma_{c1})$.

- P.5  What means “a secondary expression for the ratio of between-class variance and within-class variance.”?

- P.6  The authors wrote that “Regarding equation 9, the ratio of the Euclidean distance between the class mean vectors and the between-class variance is supported to be constant.” Why can it be supported to be constant? The feature transformation changes the distance of the mean vectors of each class.

- Abstract; “the same performance can be obtained “. The performance is not completely the same as fine-tuning nor meta-learning.

- Variance-normalization of Eqs.(12) and (13) are not clear. Is the different variance calculated for each class, or is the variance common for all samples?

- For covariance matrix, $(\phi(x)-\mu_c)^T( \phi(x) - \mu_c )$ seems to be $(\phi(x) - \mu_c) (\phi(x) - \mu_c)^T$ and $(\mu_c - \mu)^T(\mu_c - \mu)$ seems to be $(\mu_c - \mu)(\mu_c - \mu)^T$.

- p.12 the last row: $\lambda \in R^D$ seems to be $\lambda \in R^1$.



**Summary Of The Paper:**

This paper analyzes how a prototype classifier works well without fine-tuning and meta-learning. The feature is trained with cross-entropy loss with a linear projection layer. In testing, the classifier is a Prototype classifier.

This paper derives a novel generalization bound for the prototypical network and shows that focusing on the variance of the norm of a feature vector can improve performance.
The proposed upper bound is a modification of (Cao et al., 2020), and it does not require the features distributed on any specific distribution. Each covariance matrix does not have to be the same among classes.

The authors experimentally investigated several normalization methods (L2-N, V-N, LDA, EST, EST-L2-N) for minimizing the variance of the norm.


**Summary Of The Review:**

The theoretical analysis of the variance of the norm to impact the bound of the prototypical classifier is interesting. However, the descriptions seem to contain errors, and experimental validation is not satisfactory.

==
Post rebuttal
==

My concerns have been solved. I have slightly increased the score accordingly.

---

> ### Author Response · Authors · 2021-11-17
> **Response to Reviewer krD7**
>
> We would like to thank reviewer krD7 for providing valuable feedback and raising interesting questions which we answer below.
>
> **Q1. The L2-N is applied only to EST. Why L2-N is not applied to V-N and LDA.**
>
> **A1** Since LDA and V-N were inferior to L2-N, we consider their combination unimportant and omitted due to the lack of space. We have uploaded the result in Appendix 6.
>
> **Q2. It is not clear if B@CL2 and B+@CL2 use a prototypical classifier, or fine-tuning is conducted.**
>
> **A2** We have modified the next sentence in Section 4.2 as follows.
>
> Before: In that study they transformed the features so that the mean of all features to be origin before $L_2$-normalization.
>
> After: In that study they transformed the features so that the mean of all features to be origin before $L_2$-normalization and then use a prototype classifier without fine-tuning a linear classifier.
>
> **Q3. The discussion of Sec.4.4 is not completely validated. It claims that the L2-norm normalization less influences  $\mathrm{Tr}(\Sigma) $ than $\mathrm{Tr}(\Sigma_c)$. These values before and after the normalization also should be compared.**
>
> **A3** We have uploaded the results in Figure4 in Appendix A.7. When the backbone is ResNet12 trained on miniImagenet (this is the same settings with the model used in figure 3.), the value $\mathrm{Tr}(\Sigma_c) / \mathrm{Tr}(\Sigma)$ (the lower the value is, the lower the bound is.) changes from 4.51 to 4.40 and marginally reduces the ratio. However, in other settings, L2-norm can marginally increase the ratio or have no influence on the ratio.
>
> **Q4. Gaussian distribution are used when expanding the results of one-sided Chebyshev's inequality, eg., Eq.(22). Also, p.15 Eq.(29), the first line to the second line, it seems to be assuming $\mathrm{Tr}(\Sigma_{c_2})=\mathrm{Tr}(\Sigma_{c_1})$.**
>
> **A4** Eq. (22) defines the variance of the features in the class conditional distribution and does not assume a Gaussian distribution. We take the expectation of the class mean vectors In Eq. (29), $\mu_{c_1}$ and $\mu_{c_2}$, at $c_1$ and $c_2$, so it becomes $\mathrm{Tr}(\Sigma)$ and we do not assume $\mathrm{Tr}(\sigma_{c_2})=\mathrm{Tr}(\Sigma_{c_1})$. $\mathrm{Tr}(\Sigma_{c_1})$ and $\mathrm{Tr}(\Sigma_{c_2})$ only appear in the class conditional distribution, so it does not appear here.
>
> **Q5. P.5 What means “a secondary expression for the ratio of between-class variance and within-class variance.”?**
>
> **A5** When $r = \frac{\mathrm{Tr}(\Sigma_c)}{\mathrm{Tr}(\Sigma)}$, the term “$\mathrm{V}_wit(\Sigma_\tau, \Sigma, \mu)$” is expressed in a quadratic formulation of $r$, i.e, $ar^2 + br + c$. ($a,b,c$ is coefficient and constant values not related to $r$)
>
> **Q6. Why can the ratio of the Euclidean distance between the class mean vectors and the between-class variance be supported to be constant? The feature transformation changes the distance of the mean vectors of each class.**
>
> **A6** When $r = \frac{\mathrm{E}{\mathrm{dist}^2_{L2}}(\mu_{c_1}, \mu_{c_2})}{\mathrm{Tr}(\Sigma)^2}$, $r$ is equivalent to the ratio of the between-class variance to the Euclidean distance between the class mean vectors. When we minimize the Euclidean distance between the class mean vectors, the between-class variance also reduces in the same ratio with the distance. Therefore we assume the r to be constant.
>
> **Q7. Abstract; “the same performance can be obtained “. The performance is not completely the same as fine-tuning nor meta-learning.**
>
> **A7** We modified it as ``comparable performance can be obtained’’.
> In some experiments, we exceeded the performance of the meta-learning-based and fine-tuning-based methods;however,  we were interested in achieving the same level of performance, i.e., our goal is not to exceed the performance of the meta-learning-based method and the fine-tuning-based method but to achieve at least the same level of performance. Thus,  we use the word  "same" in the abstract.
> However, as you have pointed out, the statement “the same performance can be obtained” may be confusing.
>
>
> **Q8. Variance-normalization of Eqs.(12) and (13) are not clear. Is the different variance calculated for each class, or is the variance common for all samples?**
>
> **A8** We calculated the variance for each class. However, some dimensions of the variances can be zero due to the ReLU activation function. Thus,  we replaced the zero-value dimension with the corresponding dimension of variance calculated for the whole class.
>
> **Q9. For covariance matrix, $(\phi(x) - \mu_c)^\top (\phi(x) - \mu_c)$ seems to be $(\phi(x) - \mu_c)(\phi(x) - \mu_c)^\top$ and
> $(\mu_c - \mu)^\top(\mu_c - \mu)$ seems to be $(\mu_c - \mu)(\mu_c - \mu)^\top$.
> p.12 the last row: $\lambda \in \mathbb{R}^d$ seems to be $\lambda \in \mathbb{R}^1$**
>
> **A9** Thank you for pointing that out. We have modified the term.

---

> > ### Comment · Reviewer_krD7 · 2021-11-25
> > **Post rebuttal comment**
> >
> > The authors addressed most of my concerns in the initial reviews. Experiments have been improved. I think the theoretical analysis when training the features on softmax loss and Prototypical Networks on test time is valuable while reviewe1 concerned the significance.
> > I think the writing of this paper can be improved, though it is not seriously bad.
> > Due to that, I would like to keep the initial rating. The remaining concerns are as follows:
> >
> > >A.5   When $r = \frac{\rm Tr(\Sigma_c)}{\rm Tr(\Sigma)}$,
> > the term $V_{wit}(\Sigma_r, \Sigma, \mu) $ is expressed in a quadratic formulation of r, ...
> >
> >  $\Sigma_c$ is class $c$’s covariance matrix.
> > Does $\Sigma_c$ mean $\Sigma_{\tau}$ and  $V_{wit}(\Sigma_r, \Sigma, \mu)$ mean $\frac{V_{wit}(\Sigma_{\tau}, \Sigma, \mu )}{\rm Tr (\Sigma) } $?
> >
> > P.5 wrote that "The term $V_{wit}(\Sigma_\tau, \Sigma, \mu)$ is correlated with the within-class variance because
> > $\frac{V_{wit}(\Sigma_{\tau}, \Sigma, \mu )}{\rm Tr (\Sigma) }$  is a secondary expression for the ratio of between-class variance and within-class variance. ”
> >
> > If so, could the authors show the $a, b, c$ explicitly?  I think they include between class variance $\Sigma$. Thus, the term $V_{wit}$ includes both within-class and between-class variance. This should be justified.
> >
> > > Sec.3.4/A.7 etc: ''the ratio of between-class variance to the within class variance’’
> >
> > Ratio of A to B means A/B. It should be ''the ratio of the within class variance to the between-class variance”
> >
> >
> > > A.7 “We calculated the variance for each class”.
> >
> > Eq.(16) is still confusing. The variance is calculated from the first input.
> >
> > >Typos
> >
> > P5.  Eq.(7) $\Sigma_{c_i}$
> >
> > P.5  Remark $\phi(S_y)$
> >
> > Fig.2 $\phi(x_c)$, $V_{with}(\Sigma_c, …)$

---

> > > ### Author Response · Authors · 2021-11-25
> > > **Response to Post rebuttal comment of Reviewer krD7 Part2**
> > >
> > > >**Q. A.7 “We calculated the variance for each class”.
> > > Eq.(16) is still confusing. The variance is calculated from the first input.**
> > >
> > > Thank you for pointing that out. Since the expression is confusing, we will revise the Eq.(16) as follows. We express the variance of input $\boldsymbol{x}$ with class-conditioned distribution $\mathcal{D_c}$, where $c$ is the class of $\boldsymbol{x}$.
> > >
> > > Before:
> > > >$$
> > > \\psi_{\mathrm{maha}} (\\phi(\\boldsymbol{x}_1), \\phi(\\boldsymbol{x}_2)) = \\sum_d^D \frac{\\left( \\phi(\\boldsymbol{x}_1)^{(d)}-\\phi(\\boldsymbol{x}_2)^{(d)}\\right)^2}{\\overline{\\mathrm{Var}[\\phi(\\boldsymbol{x}_1)^{(d)}]}},
> > > $$
> > >
> > > After:
> > >
> > > >$$
> > > \\psi_{\\mathrm{maha}}(\\phi(\\boldsymbol{x}), \\phi(\\boldsymbol{x}_q)) = \\sum_d^D \mathrm{D_m}^{(d)}
> > > $$
> > >
> > > >where
> > >
> > > >$$
> > > \mathrm{D_m}^{(d)} = \\frac{\\left(\\phi(\\boldsymbol{x})^{(d)}-\\phi(\\boldsymbol{x}_{q})^{(d)}\\right)^2} {\\overline{\\mathrm{Var|\boldsymbol{x}\sim \mathcal{D}_c}[\\phi(\\boldsymbol{x})^{(d)}]}}.
> > > $$
> > >
> > > >and $\\mathrm{Var|\boldsymbol{x}\sim \mathcal{D}_c}$ means the variance over $x$ distributed as $\mathcal{D}_c$.
> > >
> > > The first input indicates the samples in the support set and the second input indicates the samples in the query set. The class of the second input is not available.
> > >
> > > >**P5. Eq.(7) $\Sigma_{c_i}$, P5. Remark $\phi(S_y)$, Fig.2 $\phi(\boldsymbol{x_c}), V_{wit}(\Sigma_c, ...)$**
> > >
> > > Thank you for pointing that out. We will revise as follows.
> > > >**A. P5. Eq.(7) $\Sigma_{c_i}$**
> > >
> > > Before: $\Sigma_{c_i}$
> > >
> > > After: 	$\Sigma_c$
> > >
> > > >**P5. Remark $\phi(S_y)$**
> > > Before: $\phi(S_y)$
> > >
> > > After: $\phi(\boldsymbol{x})$
> > >
> > > >**Fig.2**
> > >
> > > Before: $\phi(\boldsymbol{x_c}), \mathrm{V_{wit}(\Sigma_c, \Sigma, \mu)}$
> > >
> > > After: $\phi(\boldsymbol{x}), \mathrm{V_{wit}(\Sigma_\tau, \Sigma, \mu)}$

---

> > > > ### Comment · Reviewer_krD7 · 2021-11-26
> > > > **Re: Response to Post rebuttal comment of Reviewer krD7**
> > > >
> > > > The rebuttal solved my concerns. I have updated the score accordingly.

---

> > > ### Author Response · Authors · 2021-11-25
> > > **Response to Post rebuttal comment of Reviewer krD7 Part1**
> > >
> > > We are really appreciative of your comment on our response.
> > > Since the paper revision period in the rolling discussion has expired,
> > > let us report on the revised contents in this thread.
> > >
> > > > **I think the writing of this paper can be improved, though it is not seriously bad.**
> > >
> > > We hope that the following revise will dispel your concerns.
> > >
> > > > **$\Sigma_c$ is class $c$’s covariance matrix. Does $\Sigma_c$ mean $\Sigma_\tau$ and $\mathrm{V_{wit}}(\Sigma_r,\Sigma,\mu)$ mean $\frac{\mathrm{V_{wit}}(\Sigma_\tau,\Sigma,\mu)}{\mathrm{Tr}(\Sigma)}$?**
> > >
> > > As you have pointed out, $\Sigma_c$ is $\Sigma_\tau$. Regarding $\mathrm{V_{wit}}(\Sigma_r,\Sigma,\mu)$,  it  means $\frac{\mathrm{V_{wit}}(\Sigma_\tau,\Sigma,\mu)}{\left(\mathrm{Tr}(\Sigma)\right)^2}$ as I will mention in the next.
> > >
> > >
> > > >**P.5 wrote that "The term $\mathrm{V_{wit}}(\Sigma_r,\Sigma,\mu)$ is correlated with the within-class variance because
> > > $\frac{\mathrm{V_{wit}}(\Sigma_\tau,\Sigma,\mu)}{\mathrm{Tr}(\Sigma)}$ is a secondary expression for the ratio of between-class variance and within-class variance. ” If so, could the authors show the a,b,c explicitly? I think they include between class variance $\Sigma$. Thus, the term $\mathrm{Vwit}$ includes both within-class and between-class variance. This should be justified.**
> > >
> > > As you have pointed out, the expression is confusing. So we will revise the expression as follows.
> > >
> > > Before: The term $\mathrm{V_{wit}}(\Sigma_r,\Sigma,\mu)$ is correlated with the within-class variance because
> > > $\frac{\mathrm{V_{wit}}(\Sigma_\tau,\Sigma,\mu)}{\mathrm{Tr}(\Sigma)}$ is a secondary expression for the ratio of between-class variance and within-class variance.
> > >
> > > After: The term $\frac{\mathrm{V_{wit}}(\Sigma_r,\Sigma,\mu)}{\left(\mathrm{Tr(\Sigma)}\right)^2}$ is monotonically increasing with respect to $\mathrm{Tr}(\Sigma)$ and monotonically decreasing with respect to $\mathrm{Tr}(\Sigma)$. This indicates that the term $\frac{\mathrm{V_{wit}}(\Sigma_r,\Sigma,\mu)}{\left(\mathrm{Tr(\Sigma)}\right)^2}$ is monotonically decreasing with respect to the ratio of the within-class variance to the between-class variance. Thus the term $\frac{\mathrm{V_{wit}}(\Sigma_r,\Sigma,\mu)}{\left(\mathrm{Tr(\Sigma)}\right)^2}$ is affected by the ratio of the within-class variance to the between-class variance.
> > >
> > > We revise the expression as above because
> > > $\frac{\mathrm{V_{wit}}(\Sigma_\tau,\Sigma,\mu)}{\left(\mathrm{Tr}(\Sigma)\right)^2}$ can be written as follows:
> > >
> > > $$
> > >     \frac{\mathrm{V_{wit}}(\Sigma_\tau,\Sigma,\mu)}{\left(\mathrm{Tr}(\Sigma)\right)^2} = \frac{8}{K} \left(\frac{\mathrm{Tr}(\Sigma_\tau)}{\mathrm{Tr}(\Sigma)}\right)^2 + \frac{8}{K}\frac{\mathrm{Tr}(\Sigma_\tau)}{\mathrm{Tr}(\Sigma)} + \frac{8}{K}\frac{\mathrm{Tr}(\Sigma_\tau)}{\left(\mathrm{Tr}(\Sigma)\right)^2}\mu^\top\mu+4\left(1+\frac{\mu^\top \mu}{\mathrm{Tr}(\Sigma)}\right)^2
> > > $$
> > >
> > > This equation indicates that the term $\frac{\mathrm{V_{wit}}(\Sigma_\tau,\Sigma,\mu)}{\mathrm{Tr}(\Sigma)}$ is monotonically increasing w.r.t. $\mathrm{Tr}(\Sigma_\tau)$ and monotonically decreasing w.r.t. $\mathrm{Tr}(\Sigma)$.
> > > Thus, we used the expression
> > > "The term $\mathrm{V_{wit}}(\Sigma_r,\Sigma,\mu)$ is correlated with the within-class variance because
> > > $\frac{\mathrm{V_{wit}}(\Sigma_\tau,\Sigma,\mu)}{\mathrm{Tr}(\Sigma)}$ is a secondary expression for the ratio of between-class variance and within-class variance.".
> > >
> > > >**Sec.3.4/A.7 etc: ''the ratio of between-class variance to the within class variance’’
> > > Ratio of A to B means A/B. It should be ''the ratio of the within class variance to the between-class variance”**
> > >
> > > Thank you for pointing that out. We will revise the expressions in our paper as follows.
> > > Before:
> > > - `the ratio of between-class variance to the within-class variance',
> > > - `increasing the ratio of the between-class variance to the within-class variance' (Section A.3)
> > >
> > > After:
> > > - `the ratio of the within-class variance to the between-class variance',
> > > - `decreasing the ratio of the within-class variance to the between-class variance'

---

### Official Review · Reviewer_M6Y5 · 2021-11-05

**Correctness:** 4
**Technical Novelty And Significance:** 3
**Empirical Novelty And Significance:** 2
**Recommendation:** 5
**Confidence:** 4

**Main Review:**

The topic studied in the paper is relevant to the few-shot learning research community, and the prospect of applying prototypical classifiers to a wider range of pre-trained feature extractors is appealing. The related work section is thorough and does a good job of contextualizing the proposed approach.

My expertise is limited in terms of evaluating the correctness of the derived bound; I will leave this to other reviewers and take it at face value in my review. The main issue I have with the submission is clarity. In several instances the notation tripped me up, or the writing was imprecise, or some experimental details were unclear:
- The term "linear classifier with fine-tunining" doesn't make sense to me. I know this formulation has been used in the literature before, but calling it "fine-tuning" is incorrect when the backbone is frozen and a new classifier is learned from scratch.
- Are there typos in the joint probability distributions defined in Section 3.1's third line? In both cases, the index "i" in the product is not used in the expression.
- In Equation 2, should c ~ \tau read as c_1, c_2 ~ \tau?
- Where is Tr(\Sigma_c) defined? It's used in Section 4.4, but the closest definition I could find was the within-class variance \Sigma_\tau discussed in item 2 at the bottom of page 5.
- Since the drawbacks of Cao et al.'s bound are discussed in this work, it would be good to show the bound in the submission. It would avoid having to pull up Cao et al.'s paper and put it side by side with the submission just to compare the bounds.
- Comparing notations with Cao et al., I find their choice of notation and the way they introduce it much clearer, and I wish this submission adopted their notation (with attribution, of course).

I also have some reservations about the experiments:
- Can the authors expand on why the center loss and affinity loss are left out of the comparisons? Is it because they need to be applied during pre-training? If so, it should be made clearer.
- Again, the following is common to a lot of few-shot classification papers, but given the tiered-ImageNet results, what additional evaluation signal is expected from the mini-ImageNet results? Isn't it redundant? The same can be said about FC100 vs CIFAR-FS.
- How is Baseline++ trained in the re-implementation? Section 4.2 mentions the normalization of the projection layer and features as a difference between it and Baseline, but omits the cosine classifier.
- The gap between B+@FT and B@FT is surprisingly small, given the improvements of Baseline++ over Baseline reported by Chen et al. Do the authors have an explanation? Is it possible that Baseline++ was not pre-trained properly in their re-implementation? In particular, Baseline++ scales the normalized weight vectors independently for each class, which is not clearly mentioned in Chen et al.'s paper, but is necessary for good performance. Is that detail also present in the re-implementation?
- The submission goes beyond standard practice and uses a principled criterion for bolding table entries by comparing 95% confidence intervals, which is a good thing that I would like to see in more submissions. I would like to see those 95% confidence intervals in the Appendix, though, so that future papers can also compare to those confidence intervals.

In terms of significance, I would argue that "comparable with ProtoNet and the fine-tuning approach" is not that high a bar. I like the premise of being able to plug a prototypical classifier into any pre-trained model, and I feel that it could have been better exploited in the submission, for instance by using a larger-scale pre-trained classifier out of the box and achieving even better few-shot classification accuracies. I would say however that I am not entirely sold on a prototypical classifier having more desirable properties than a linear classifier or an SVM (like is done with MetaOptNet (Lee et al., 2019)). The paper justifies this through the lens of hyperparameter tuning, but work like Big Transfer (Kolesnikov et al., 2020) shows that heuristics such as BiT-HyperRule can be designed to avoid hyperparameter sweeps.

Additional questions/comments:
- The abstract alternates between present and past verb tense. It would flow better if it was consistent in its verb tense use.
- Since the variance-normalization transformation is described as a variation of the Mahalanobis distance with a diagonal covariance matrix, it would be relevant to cite Bateni et al. (2020)'s Simple CNAPs approach, which uses a Mahalanobis distance-based classification criterion.

References:
- Bateni, P., Goyal, R., Masrani, V., Wood, F., & Sigal, L. (2020). Improved few-shot visual classification. In CVPR.
- Kolesnikov, A., Beyer, L., Zhai, X., Puigcerver, J., Yung, J., Gelly, S., & Houlsby, N. (2020). Big Transfer (BiT): General visual representation learning. In ECCV.
- Lee, K., Maji, S., Ravichandran, A., & Soatto, S. (2019). Meta-learning with differentiable convex optimization. In CVPR.

**Summary Of The Paper:**

The submission introduces a generalization bound for Prototypical Networks that does not depend on assumptions on the class-conditional distributions. This bound decreases as the variance in norm of the feature vectors decreases, and the authors empirically investigate five feature transformation approaches that are meant to lower this variance: L2-normalization (L2-norm), variance-normalization, Linear Discriminant Analysis (LDA), Embedding Space Transformation (EST), and EST+L2-norm.

Experimental results are reported on the mini-ImageNet, tiered-ImageNet, CIFAR-FS, and FC100 benchmarks in the 1-shot and 5-shot settings, using ResNet-12 or ResNet-18 as the backbone architecture. The five feature transformation approaches are applied on top of the Baseline and Baseline++ learners before using the extracted features to build a prototypical classifier. The submission compares against the following:
- Prototypical Networks
- Baseline and Baseline++ with a linear classifier on top of the extracted features
- Baseline and Baseline++ with Wang et al.'s centering+L2-norm approach on top of the extracted features.
- Baseline and Baseline++ with no additional feature transformation and a prototypical classifier on top of the extracted features.

The submission concludes that EST+L2-norm performs best, and that its performance-boosting effect decreases as the number of examples per class increases.

**Summary Of The Review:**

The paper studies an interesting topic but lacks clarity. I have concerns with the experimental setup, and with how the paper justifies the significance of its proposed approach.


---

**POST-REBUTTAL**: Most of my concerns are addressed, but I remain concerned with the submission's significance (see discussion thread).

---

> ### Author Response · Authors · 2021-11-17
> **Response to Reviewer M6Y5 part1**
>
> We would like to thank reviewer M6Y5 for providing valuable feedback and raising interesting questions which we answer below.
>
>  **Q1. "Fine-tuning" is incorrect when the backbone is frozen and a new classifier is learned from scratch.**
>
> **A1** We used this term in line with the following previous few-shot learning studies.
> - Chen et al. (2019) uses “fine-tuning” for the test-phase of  Baseline++ and Baseline.
> - Liu et al. (2020) says "In the fine-tuning stage, as there are only few labeled samples  for training (e.g. 5-way 1-shot learning only contains 5 training samples), we follow to fix the parameters of the backbone, and only train a new classifier from scratch"
> - Wang et al. (2020) says "We find that fine-tuning only the last layer of existing detectors on rare classes is crucial to the few-shot object detection task."
>
> Furthermore, "fine-tuning" is used not only to indicate to update all parameters of the models but also to update part of the parameters in transfer learning literature. For example,  Guo et al. (2019) use the "fine-tuning" as follows: "If the target dataset is small and the number of parameters is huge, fine-tuning the whole network may result in overfitting. Alternatively, the last few layers of the deep network can be fine-tuned while freezing the parameters of the remaining initial layers to their pre-trained values." and Tajbakhsh et al. (2016) says "Therefore, fine-tuning the last few layers is usually sufficient for transfer learning".
> However, in self-supervised learning literature (Chen et al., 2020), they used the term “linear evaluation” for training a new classifier is learned from scratch with a frozen backbone. Would “linear evaluation”  be better?
>
> **References**
> - Chen, W., Liu, Y., Kira, Z., Wang,Y., and Huang, J. (2019)  A Closer Look at Few-shot Classification. In ICLR
> - Liu, B., Cao, Y., Lin, Y., Li, Q., Zhang, Z., Long, M., and Hu, H. (2020) Negative Margin Matters: Understanding Margin in Few-shot Classification. In ECCV
> - Wang, X.,  Huang, T., Darrell, T., Gonzalez, J., and Yu, F. (2020) Frustratingly Simple Few-Shot Object Detection. In ICML
> - Guo, Y., Shi, H., Kumar, A., Grauman, K., Rosing, T., and Feris, R. (2019) SpotTune: Transfer Learning through Adaptive Fine-tuning. In CVPR
> - Tajbakhsh, N., Shin, J., Gurudu, S., Hurst, T., Kendall, C., Gotway, M., and Liang, J. (2016) Convolutional Neural Networks for Medical Image Analysis: Full Training or Fine Tuning? In IEEE transactions on medical imaging.
> - Chen, T.,  Kornblith, T., Norouzi, M., and Hinton, G. 1. (2020). A Simple Framework for Contrastive Learning of Visual Representations. In ICML.
>
> **Q2. Are there typos in the two joint probability distributions defined in Section 3.1's third line?**
>
> **A2** For the first one $D_y^{\otimes n}$, ‘$i$’ does not appear because $D_y^{\otimes n}$ represents the joint distribution of n data sampled from the same probability distribution Dy. We fixed the typo in $D^{\otimes nk}$, thank you for pointing that out.
>
> **Q3. In Equation 2, should $c \sim \tau$ read as $c_1, c_2 \sim \tau$?**
>
> **A3** We are considering the $N$-class classification in Equation 2, and the class of the query data is $c$; thus, we can leave it as it is. Since $c_2$ appears as a result of decomposition of $S\sim D^{\otimes nk}$, we took the expectation values for $c_1$ and $c_2$ as in Equation 9.
>
> **Q4. Where is $\mathrm{Tr}(\Sigma_c)$ defined?**
>
> **A4** As described in Section 3.3, we define $\Sigma_c$ as the covariance matrix of class $c$’s extracted features. As you have pointed out, it was difficult to understand, so I added the explanation in Section 4.4 .
>
> **Q5. It would be good to show the Cao’s bound in the submission.**
> **A5** As described in Section 3.2 and Section A.1, we show the Cao’s bound in Section A.1. We show the bound in Appendix due to a lack of space.
>
> **Q6. Comparing notations with Cao et al., I find their choice of notation and the way they introduce it much clearer, and I wish this submission adopted their notation**
>
> **A6** Our notations are designed to make it easier for machine learning researchers to read. By convention, generalization bounds are typically analyzed for loss or risk. The orientation of the inequality is often confusing, and it is easier to intuitively understand the statement and proof when it is according to the convention.
> Also, since it was a little difficult to understand the expected calculation in the existing paper, I decided to write it in separate sections so that we can understand which variables depend on other variables, where we are considering class conditional probability, and how the expectation value can be decomposed.
> While Cao et al. used the variables $a$ and $b$ for the class variables, I used $c$ and the subscript $c_j$ because $c$ is often used as a convention, and the $a$ and $b$ notations are not convenient when the number of classes increases to $N$.

---

> ### Author Response · Authors · 2021-11-17
> **Response to Reviewer M6Y5 part2**
>
> We would like to thank reviewer M6Y5 for providing valuable feedback and raising interesting questions which we answer below. This is part2 of our answer.
>
> **Q7. Can the authors expand on why the center loss and affinity loss are left out of the comparisons? Is it because they need to be applied during pre-training? If so, it should be made clearer.**
>
> **A7** Just as you have pointed out, we did not use the losses because we wanted to use the same loss function with previous work, and our goal is not to achieve SOTA but through the comparison with existing studies showing that we can achieve the same level of performance without fine-tuning.
>
> **Q8. What additional evaluation signal is expected from the mini-ImageNet results or the FC100 results? Isn't it redundant?**
>
> **A8** The difference between the dataset is the class splitting protocol. The miniImagenet and CIFARFS are divided randomly, and tieredImagenet and fc100 are divided on the basis of the super-category. That is, the similarity of the classes between training and testing is different. The experimental results and accuracy are quite different between cifar100 and FC100, and L2-N alone loses to EST alone. We design the experiments in line with the following previous studies.
>
> - Cao, T., Law, M., and Fidler, S. (2020) A theoretical analysis of the number of shots in few-shot learning. In ICLR.
>     - miniImagenet and tieredImagenet
> - Tian, Y., Wang, Y., , Krishnan, D., Tenenbaum, J., and Isola, P. (2020) Rethinking Few-Shot Image Classification: a Good Embedding Is All You Need? In ECCV
>     - minimagenet and tieredImagenet, CIFARFS and FC100
> - Rodr´ıguez, P., Laradji, I., Drouin, A., and Lacoste, A. (2020) Embedding Propagation: Smoother Manifold for Few-Shot Classification In ECCV
>     - miniImagenet and tieredImagenet
> - Ziko, I., Dolz, J., Granger, E., and Ayed, B. (2020) Laplacian Regularized Few-Shot Learning. In ICML
>     - miniImagenet and tieredImagenet
> - Yoon, S., Seo, J., and Moon, J. TapNet: Neural Network Augmented with Task-Adaptive Projection for Few-Shot Learning.
>     - miniImagenet and tieredImagenet
>
> **Q9. How is Baseline++ trained in the re-implementation? Section 4.2 mentions the normalization of the projection layer and features as a difference between it and Baseline, but omits the cosine classifier.**
>
> **A9** As described in section 4.2, we normalize the projection layer and the features. We describe the cosine classifier in this way.
>
> **Q10. What makes the gap between the authors’ reimplementation of Basline++ and Chen’s Baseline++?**
>
> **A10** We also scales the normalized weight vectors independently for each class just like Chen’s work. We evaluate the model every 5 epochs, not every epoch, and we perform early stopping when the validation score does not improve for 50 epochs. Therefore, a performance difference of about $1.5$\% is considered to appear.
> This may be the reason why the baseline performed $2$\% better than the original paper, preventing overfitting to the validation data.
>
> **Q11. Can the authors add $95$\% confidence intervals of performance results in Appendix?**
>
> **A11** We added the intervals in Appendix 6.
>
> **Q12.  A prototypical classifier does not have more desirable properties than a linear classifier or an SVM (like is done with MetaOptNet (Lee et al., 2019)). The paper justifies this through the lens of hyperparameter tuning, but work like Big Transfer (Kolesnikov et al., 2020) shows that heuristics such as BiT-HyperRule can be designed to avoid hyperparameter sweeps.**
>
> **A12** A prototype classifier has advantages over training new linear classifiers other than hyperparameter tuning. That is, a prototype classifier does not require the models to be retrained when the number of classes increases during transfer learning with Few-shot. This can enable the model to quickly adapt to new classes. However, MetaOptNet requires the training of SVM when new classes appear and it is difficult to quickly adapt to new classes. Besides, a linear classifier or an SVM (SVMs)  requires training at test time, so there will be a time lag before it can be operationalized. Also, while prototype classifiers require only the memory of the data store, SVMs require the memory of the machine during training, making it difficult to run on edge devices.
>
> Regarding the Bit-hyper rule, it is a heuristic and there is no theoretical guarantee. Therefore, we do not know if it still works well when the backbones are changed. Also, they use ImageNet and JFT for the pre-training data set, but we don't know if it will work when we change the dataset in the pre-training phase.
> .
>
>
>
> **Q13. The abstract alternates between present and past verb tense**
> **A13** We have fixed the verb tense to present. Thank you for pointing this out.
>
> **Q14. It would be relevant to Bateni et al. (2020)'s Simple CNAPs approach.**
>
> **A14** We have cited the work.

---

> > ### Comment · Reviewer_M6Y5 · 2021-11-19
> > **Updated review**
> >
> > Thank you for your response. Most of my concerns are addressed.
> >
> > RE: "fine-tuning" vs "linear evaluation", as mentioned in my review, I am aware that the "fine-tuning" terminology is used in previous work, and I can't fault this submission for following convention, but this is one of those instances where departing from the established convention for clarity reasons outweighs following it for consistency reasons. I do think "linear evaluation" would be better.
> >
> > My one remaining concern is significance. I am still not convinced that the differences between Prototypical Networks and other kinds of linear classifiers matter in practice. Training an SVM is arguably "quick" when compared with fine-tuning the entire backbone, and it's unclear to me whether Prototypical Networks' "quicker" adaptation is that big an advantage.
> >
> > Regarding hyperparameter selection heuristics and their lack of theoretical guarantees, the same can be said of Prototypical Networks. The unimodal, isotropic modeling assumption makes it so that a closed-form solution exists for the adaptation step on the support set, but the extent to which the backbone "fulfills that prophecy" and outputs normally-distributed embeddings very much depends on how the test data distribution aligns with the pre-training data distribution (see e.g. Allen et al. (2019) for a discussion on that topic).
> >
> > References:
> >
> > Allen, K., Shelhamer, E., Shin, H., & Tenenbaum, J. (2019). Infinite mixture prototypes for few-shot learning. In International Conference on Machine Learning.

---

> > > ### Author Response · Authors · 2021-11-21
> > > **Response to Updated Review of Reviewer M6Y5 Part2**
> > >
> > > >**Regarding hyperparameter selection heuristics and their lack of theoretical guarantees, the same can be said of Prototypical Networks. The unimodal, isotropic modeling assumption makes it so that a closed-form solution exists for the adaptation step on the support set, but the extent to which the backbone "fulfills that prophecy" and outputs normally-distributed embeddings very much depends on how the test data distribution aligns with the pre-training data distribution.**
> > >
> > >
> > > As you have pointed out, ProtoNet is based on isotropic modeling assumption. However,
> > > multi-modal is caused by how the data is annotated. Since the number of a support set is small in few-shot settings (for example, in 5-way 5-shot, the number of labeled data is 25), we can easily check the labels and re-label them so that they don't become multi-modal. This relabeling can be enough for practical use in few-shot settings. Therefore, a prototype classifier still seems to be a reasonable method for few-shot learning. For problems that are difficult to check in the first place, other methods can be used. Therefore, it would be practically important to theoretically analyze a simple method that should be done first,  clarify its nature and limitations, and improve its performance.
> > >
> > > By the way, based on your point and our theory, we noticed that the following can be inferred. As described in our paper, it is known that when the cross entropy loss is used, the data embedded in the feature space extends in the direction of increasing the variance of the norm for each class. If the multimodality is shaped to have a large variance in the direction of the norm for each class mean vectors, reducing the variance of the norm may improve the multimodality problem in the feature space. This may be the reason for the increased performance as shown in the experimental results. We would like to leave this reasoning a future work. Thank you for your many useful comments.

---

> > > > ### Author Response · Authors · 2021-11-26
> > > > **On the importance of further research on prototype classifiers**
> > > >
> > > > Thank you for your repeated discussions with us.
> > > > We would appreciate your opinion on whether our recent feedback convinces you of the importance of further research on prototype classifiers. Your feedback so far has been valuable to us.
> > > > We believe that clarifying the differences between your opinion and ours will help us to improve our paper.

---

> > > > > ### Comment · Reviewer_M6Y5 · 2021-11-29
> > > > > **Response**
> > > > >
> > > > > Thank you for your feedback. When evaluating the impact potential I see for the submission, I'm still hesitant. In some ways, work like Dvornik et al.'s SUR and Chen et al.'s New Meta-Baseline already propose solutions to the problem of training good nearest-centroid classifiers. The former does so using a cosine-based classifier, and the latter adopts an "episodic meta-fine-tuning" strategy which alleviates the cost of meta-training. I'm still not sold on the idea that the submission's proposed solution is clearly better.
> > > > >
> > > > > At the same time, I don't think there is anything fundamentally wrong with the submission, so I would not be upset if it was accepted. I will keep my score unchanged and let the AC make their final decision in light of these points.

---

> > > ### Author Response · Authors · 2021-11-21
> > > **Response to Updated Review of Reviewer M6Y5 Part1**
> > >
> > > We are really appreciative of your reevaluating our work on the basis of our response.
> > > We would like to answer your question in more detail and would appreciate your continued discussion.
> > >
> > > >**RE: "fine-tuning" vs "linear evaluation", as mentioned in my review, I am aware that the "fine-tuning" terminology is used in previous work, and I can't fault this submission for following convention, but this is one of those instances where departing from the established convention for clarity reasons outweighs following it for consistency reasons. I do think "linear evaluation" would be better.**
> > >
> > > We have modified our paper. We changed the word “fine-tuning” to “linear evaluation” or “training a new linear classifier”.
> > >
> > >
> > > >**I am still not convinced that the differences between Prototypical Networks and other kinds of linear classifiers matter in practice. Training an SVM is arguably "quick" when compared with fine-tuning the entire backbone, and it's unclear to me whether Prototypical Networks' "quicker" adaptation is that big an advantage.**
> > >
> > >
> > > Let us make sure we have a correct understanding of your opinion. We summarize your opinion as follows.
> > >
> > > 1.  In the introduction, we described that learning in a metatest phase is problematic; however, this may not be a problem in itself.
> > > 2. A prototype classifier is a simple algorithm, and there exist better algorithms with quick adaptation;  thus improving a prototype classifier is not so important.
> > >
> > >
> > > Regarding point 1, we do not think that learning a model during a meta-test is totally negative, and it depends on the situation in real world applications.
> > > In that sense, we will revise the drawbacks of learning during metatest in the introduction (last sentence of Paragraph 3 and Paragraph 4) as follows. We would like your opinion.
> > >
> > > Before:
> > >
> > >  >however,  the fine-tuning-based approach has two drawbacks, i.e., we have to tune additional hyper-parameters for fine-tuning and to applyfine-tuning every time a new class appears, which make fast adaptation difficult in few-shot learning.
> > > To overcome the disadvantages while inherits the advantages of a linear classifier with fine tuning, we analyze how to avoid meta-learning during the training phase and the linear evaluation during the testing phase.Since a prototype classifier can be applied to any trained feature extractor and does not require model learning in the testing phase, we focus on using a prototype classifier on the testing phase and training models in a standard classification manner.
> > >
> > > After:
> > >
> > > > however,  the linear-evaluation-based approach requires retraining a linear classifier every time a new class appears.
> > > In contrast, a prototype classifier can be applied to any trained feature extractor and does not require model learning in the testing phase. Therefore, a prototype classifier can be a a practical and useful first step for few-shot learning problems. In order to avoid meta-learning during the training phase and the linear evaluation during the testing phase, we focus on using a prototype classifier on the testing phase and training models in a standard classification manner.
> > >
> > > Regarding point 2, the contribution of this study is to increase the choice of algorithms for few-shot learning. We will revise the last sentence in the conclusion as follows to make the contribution of this study clearer.
> > >
> > > Before:
> > >
> > > > We believe that this work is a significant step to understanding the behavior of a prototype classifier, which is crucial for the few-shot learning problem.
> > >
> > > After:
> > >
> > > > A prototypical classifier is expected to be  a practical and useful choice at the  first step for few-shot learning problems because of its simplicity.
> > >
> > > We theoretically analyze the property of a simple algorithm, prototype classifier, and have provided a way to improve its performance. We showed that prototype classifiers can provide comparable performance.  ‘’Simple’’ is useful in practice.
> > >
> > > This study enables the following practically useful first step.
> > > Suppose that we have to tackle an image recognition problem with very few data.
> > > The first step we do for this situation is just to normalize the features extracted by a pre-trained model you already have (may be an ImageNet pretrained model could be enough) and to use a prototype classifier on the features.
> > > Then, if the performance is enough, our work is done.
> > > If the performance is not good enough for practical use, then you can try to find a suitable method by using various other methods that are improved ones of ProtoNet.
> > > At the first step in machine learning applications, it is preferable to use a simple method in practice.
> > >
> > > We think that when we have a large amount of data, a more complex method such as SVM may be better than a prototype classifier as a discriminator, but this is not always the case for a few-shot setting. That is, since the data we can use is limited, it is possible that a simple method such as the prototype classifier will perform as well as a complex method.

---

### Decision · Program_Chairs · 2022-01-20

**Decision:**

Reject

**Comment:**

This paper explores prototype vs linear classifiers for few-shot learning. It has been found that pre-training a classifier network, followed by training a linear head can produce competitive results to meta-trained prototypical networks. A natural question therefore is whether one can directly derive prototypical classifiers from pre-trained classifiers. Naively applying this idea doesn’t work well in practice though, and this paper performs a theoretical investigation to determine why. The theory is a generalization of Cao et al., 2020, that doesn’t require assumptions on the class-conditional distributions. The theory suggests that the variance of the norm of the feature vectors plays a role, so the paper explores a few feature transformations to reduce this. It demonstrates on a few benchmark datasets that transforming the feature vectors can indeed allow us to create prototypical classifiers from pre-trained networks. As a minor quibble, the paper twice refers to “direction of the norm of class mean vectors” - This should just be the direction of the class mean vectors, right? Norm is not a direction, it’s a magnitude.

During the discussion phase, a number of questions arose, mainly around the clarity of the presentation and a request for a few additional baselines (e.g., L2 normalization combined with LDA/V-N). These points were resolved by the authors. The main outstanding issue is whether there is enough novelty/significance in the paper to merit acceptance, and on that point, the reviewers felt this is borderline. On the one hand, the theory is more general and does directly point to aspects of the feature space that can yield better generalization results. On the other hand, tricks like L2 normalization are already known, and the utility of prototype classifiers over linear classifiers like SVMs is unclear.

After careful consideration, further discussion with the reviewers, as well as the program committee, it was generally agreed that this paper does not quite meet the bar in terms of the novelty or significance of its contribution. The authors mentioned time and space benefits relative to fine-tuned classifiers in their response, and I think one way to improve the paper would be to demonstrate this advantage in a real-world application or challenging scenario.